# Mixing Configurations for Downstream Prediction

Juntang Wang [* 1]  Hao Wu [* 1]  Yihan Wang [* 1]  Dongmian Zou [1]  Shixin Xu [1]

## Abstract

Clustering-based features are widely used in machine learning, but most methods must choose a resolution—a choice that is global, fixed, and ad hoc. Recent work shows that varying the resolution parameter produces only a finite set of structurally stable partitions, known as configurations. Based on this, we introduce Configuration-Mixed Prediction (CMP), a setting where models learn to adaptively weight these configurations per sample for downstream prediction. We propose MixConfig, a plug-and-play feature augmentation module that extracts configurations from any frozen embedding and learns energy-aware mixing weights via a novel selector that jointly reasons about sample context, cluster assignments, and stability statistics. Experiments across tabular, molecular, vision, and text domains demonstrate consistent improvements over single-resolution and static baselines across diverse predictor architectures, with gains particularly pronounced in low-data regimes.

## 1. Introduction

Clustering-based features are ubiquitous in machine learning, appearing in self-supervised pretraining (Caron et al., 2018), prototype-based classification (Snell et al., 2017), and graph-based semi-supervised learning (Zhu et al., 2003). Yet, every clustering method typically requires a single resolution parameter: $k$ in $k$-means or spectral clustering (Ng et al., 2001), $\gamma$ in community detection, or a cut threshold in hierarchical methods. Consider a concrete dilemma in precision medicine: a single patient may require coarse disease categories for triage (diabetes vs. cancer) yet fine-grained biomarker clusters for personalized treatment (EGFR+ vs. KRAS+ mutations). This choice represents a trade-off: if too coarse, we lose discriminative power; if too fine, we

overfit to noise. Typically this choice is global, fixed, and ad hoc, forcing researchers to either commit to a single scale a priori or run expensive hyperparameter sweeps. The fundamental problem is that real data exhibits intrinsic multi-scale structure, and different samples can benefit from different granularities even within the same dataset.

Multi-scale structure is not an exception but the norm. Molecules exhibit hierarchical organization from atoms to functional groups to scaffolds; images span from textures to parts to objects (Lin et al., 2017; Lee et al., 2009); documents range from words to phrases to topics; social networks extend from individuals to communities to organizations. Committing to a single clustering resolution discards information at other scales—information that may be critical for specific samples or downstream tasks. While this motivates multi-resolution approaches, naively sweeping over all possible resolutions ($\gamma \in [0, \infty)$) is computationally intractable. This raises a natural question: *how many distinct, meaningful resolutions actually exist in the data?*

Recent work in resolution-based clustering provides a concrete answer: as the resolution parameter $\gamma$ varies continuously from coarse to fine, only a *finite* set of distinct, structurally stable partitions emerges (Liu et al., 2021; Pitsianis et al., 2023). These stable partitions—called *configurations*—correspond to plateaus where the clustering remains unchanged despite perturbations to $\gamma$. This finiteness is not a computational artifact but a reflection of the intrinsic multi-scale structure of the data itself. Crucially, stability indicates that these configurations capture semantically meaningful groupings rather than arbitrary parameter choices. This theoretical insight motivates a natural meta-learning question: rather than selecting a single resolution a priori, *can we learn to adaptively weight all stable configurations for downstream prediction?* Configurations are universal—extractable from any embedding space via nearest neighbor graph construction and resolution-based community detection—making them applicable across tabular, molecular, vision, and text domains.

We answer this question affirmatively with *MixConfig*, a modular feature augmentation module that learns adaptive, sample-specific configuration weights (Figure 1). Given any embedding space—whether tabular features, image representations, molecular fingerprints, or text embeddings—

---

[*]Equal contribution  [1]Zu Chongzhi Center, Duke Kunshan University, No.8 Duke Ave, Kunshan, Jiangsu, China, 215316. Correspondence to: Shixin Xu <shixin.xu@dukekunshan.edu.cn>.

*Proceedings of the 43$^{rd}$ International Conference on Machine Learning*, Seoul, South Korea. PMLR 306, 2026. Copyright 2026 by the author(s).

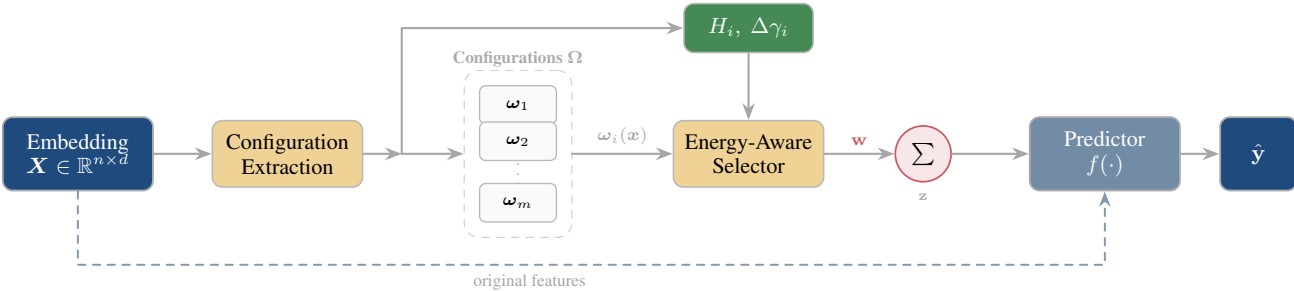

*Figure 1.* **MixConfig framework.** From an embedding $X$, we extract stable configurations $\Omega$ and stability statistics $(H_i, \Delta\gamma_i)$. An energy-aware selector produces per-sample weights $\mathbf{w}$ to mix configuration features into $\mathbf{z}$, which augments the original features for any downstream predictor.

MixConfig performs two key steps: (1) extracts the finite set of configurations $\Omega = \{\boldsymbol{\omega}_1, \ldots, \boldsymbol{\omega}_m\}$ via graph-based clustering (Section 3.3), and (2) learns to weight configurations per sample using an energy-aware selector that jointly reasons about sample context, cluster assignments, and energy-based stability statistics that encode clustering quality. The augmented features can be seamlessly passed to any downstream predictor without requiring architectural changes or hyperparameter retuning. This modularity positions *MixConfig* as a general-purpose method for leveraging multi-scale structure, orthogonal to predictor choice. Our work targets both practitioners seeking improved prediction without architectural changes, and researchers investigating multi-resolution structure as inductive bias. Experiments demonstrate consistent improvements across diverse domains (Section 4).

**Contributions.**

1. **Setting Formulation.** We introduce Configuration-Mixed Prediction (CMP), a meta-learning framework that enables models to adaptively weight multiple stable clustering resolutions per sample. To our knowledge, this work is the first to formulate configuration mixing as a supervised learning setting (Section 3.2).

2. **Model-Agnostic Architecture.** We propose MixConfig, a drop-in module with a novel energy-aware selector (Section 3.3.2) that learns sample-specific configuration weights. The selector jointly reasons about (a) sample context in feature space, (b) cluster membership at each resolution, and (c) configuration quality via energy statistics as inductive bias. MixConfig works with any embedding and any predictor architecture without modifications (Section 3.3).

3. **Empirical Findings.** Through experiments across tabular, vision, molecular and text domains, we demonstrate that MixConfig consistently outperforms single-resolution and static baselines with multiple predictor families. Notably, we observe significant gains in data-

scarce regimes and validate component contributions via rigorous ablations in Section 4.

## 2. Related Work

Existing approaches generally fall into three strategies, each constrained by fundamental limitations:

*(1) Fixed-resolution clustering.* Clustering has been widely used in representation learning. DeepCluster (Caron et al., 2018) iteratively clusters CNN features as pseudo-labels, SwAV (Caron et al., 2020) contrasts cluster assignments online, and DINO (Caron et al., 2021) discovers semantic structure through self-distillation. Similarly, Prototype-based methods (Snell et al., 2017; Vinyals et al., 2016) rely on cluster centroids for classification and few-shot tasks. However, these approaches commit to a single resolution—determined by $k$ in $k$-means or an implicit prototype count—thereby discarding multi-scale information. While hyperparameter search can find a good global resolution, it cannot adapt per sample and requires expensive cross-validation.

*(2) Hierarchical methods without principled selection.* While traditional hierarchical clustering produces a dendrogram (Von Luxburg, 2007), it lacks a learnable mechanism to select or combine levels for prediction, compelling users to manually choose cut thresholds. Recent work learns hierarchical representations in neural networks (Monath et al., 2019), but still requires manual specification of hierarchy depth. Similarly, resolution-based community detection methods like Louvain (Blondel et al., 2008) and Leiden (Traag et al., 2019) require specifying $\gamma$ a priori and suffer from the well-known "resolution limit" (Fortunato, 2010), where fine-grained clusters become undetectable in large graphs. Spectral clustering (Ng et al., 2001) similarly requires choosing cluster count beforehand. The key drawback is that these methods discover hierarchies but fail to learn from data which levels matter for the task.

*(3) Fixed multi-resolution representations.* Recent advances in Matryoshka representations (Kusupati et al., 2022) learn

nested embeddings at multiple scales through a specialized training objective. For tabular data, where tree-based methods remain dominant (Grinsztajn et al., 2022), architectures like TabNet (Arik & Pfister, 2021), FT-Transformer (Gorishniy et al., 2021), SAINT (Somepalli et al., 2021), and TabPFN (Hollmann et al., 2022) use attention mechanisms (Vaswani et al., 2017) for feature selection. However, these strategies fall short when sample-level adaptation is needed. They typically rely on fixed, architecture-specific weightings rather than learning adaptive, sample-dependent selection from supervision.

Our work unifies these approaches while overcoming their limitations. We build on recent theoretical work showing that resolution-based clustering produces only a finite set of stable partitions called configurations (Liu et al., 2021; Pitsianis et al., 2023)—these correspond to plateaus where the clustering remains unchanged despite perturbations to $\gamma \in [0, \infty)$. Like hierarchical methods (approach 1), we extract multiple resolutions; like multi-resolution representations (approach 3), we encode multiple scales; but unlike both, we learn adaptive, sample-specific weights from supervision rather than using fixed global selections or architecture-specific heuristics. This positions MixConfig as a general-purpose feature augmentation method orthogonal to predictor choice—it works with tree-based methods (XGBoost, RF) where attention mechanisms cannot be applied, and with neural methods (MLP, transformers) where it provides complementary structural information.

To our knowledge, no prior work formulates configuration-mixing problems or proposes an energy-aware selector for adaptive multi-resolution prediction. While conceptually similar to Mixture-of-Experts (MoE) (Jacobs et al., 1991; Shazeer et al., 2017; Fedus et al., 2022), the resemblance is strictly superficial. MoE architectures achieve task decomposition by routing inputs to specialized trainable subnetworks via gating networks. In contrast, we weight precomputed structural representations (configurations) using fixed energy-based stability statistics as inductive bias. Our selector learns which resolution scales contribute to prediction rather than determining which expert subnetworks to activate. The closer approach is clustering-based pseudo-labeling (Xie et al., 2020); however, existing methods use single resolutions. Classical cluster ensembles (Strehl & Ghosh, 2002) also combine multiple partitions, but seek a single discrete consensus that collapses the multi-resolution information. Our key insight is that the energy statistics associated with configurations (e.g., plateau width, attraction/repulsion balance) provide a strong inductive bias for learning which scales matter, rather than treating all resolutions as arbitrary features.

Methods like DeepCluster, SwAV, and DINO learn representations by incorporating clustering into the training objective—they modify the embedding space itself. In contrast, MixConfig operates downstream: given any fixed embedding (whether from self-supervised pretraining, supervised models, or handcrafted features), it extracts configurations and learns to weight them for prediction. This makes MixConfig orthogonal to representation learning methods: one could seamlessly use DINO-pretrained embeddings as input to MixConfig, combining learned representations with adaptive multi-resolution features. We do not position MixConfig as a competitor to these methods, but as a complementary downstream module.

## 3. Methods

We first formalize Configuration-Mixed Prediction (CMP) as a setting, then present MixConfig as a solution.

### 3.1. Preliminaries: Configurations

As the resolution parameter $\gamma$ varies from coarse to fine, the optimal clustering changes only at discrete transition points, yielding a *finite* set of stable plateau partitions, called configurations. Each configuration persists over an interval $[\gamma^-, \gamma^+]$ whose width indicates stability.

Given $n$ data items with feature matrix $\boldsymbol{X} \in \mathbb{R}^{n \times d}$, hierarchical clustering seeks all "good" partitions. Each partition is represented as a vector of cluster indices $\boldsymbol{\omega} \in \{1, \ldots, |\boldsymbol{\omega}|\}^n$, where $|\boldsymbol{\omega}|$ denotes the number of clusters in that partition (varying across different partitions).

**Definition 3.1** (Partition as a cluster-assignment vector). A partition of $n$ items is represented by an assignment vector $\boldsymbol{\omega} \in \{1, \ldots, |\boldsymbol{\omega}|\}^n$, where $\omega_i$ is the cluster label of item $i$. The vector induces a set of non-empty clusters $\mathcal{C}_k(\boldsymbol{\omega}) \coloneqq \{i \in \{1, \ldots, n\} : \omega_i = k\}$ for $k = 1, \ldots, |\boldsymbol{\omega}|$.

We follow prior work for the configuration framework and its finiteness/extraction guarantees (Pitsianis et al., 2023). We construct a $k_{\mathrm{nn}}$-nearest neighbor graph $G = (V, E)$ where vertices represent samples and edges connect similar samples, enabling resolution-based community detection with parameter $\gamma \in [0, \infty)$.

Resolution-based clustering optimizes a Hamiltonian energy function (LeCun et al., 2006; Du & Mordatch, 2019) that balances attraction (grouping similar items) and repulsion (separating dissimilar items):

$$H(\boldsymbol{\omega}) = h_a + \gamma h_r \tag{1}$$

$$h_a = -\sum_{k=1}^{|\boldsymbol{\omega}|} \sum_{i<j} w_{ij} \, \mathbf{1}_{\omega_i = \omega_j = k} \quad \text{(attraction)}$$

$$h_r = \sum_{k=1}^{|\boldsymbol{\omega}|} \left( \sum_{i<j} w_{ij} \, \mathbf{1}_{\omega_i = k} \right)^2 \quad \text{(repulsion)}$$

where $w_{ij} \geq 0$ are pairwise geodesic distance ($k$NN edge weights in our case), and $|\boldsymbol{\omega}|$ denotes the number of clusters in partition $\boldsymbol{\omega}$. The attraction term rewards co-clustering similar pairs ($\omega_i = \omega_j$), while the repulsion term does not require co-clustering: following Pitsianis et al. (2023), it penalizes repulsive connections incident to items assigned to cluster $k$ (the index $j$ ranges over all items, so repulsion is not restricted to within-cluster pairs). As $\gamma$ increases, minimizing $H(\boldsymbol{\omega})$ increasingly favors partitions that separate items or regions with large repulsive mass from others, yielding the observed split transitions in $\boldsymbol{\omega}^*(\gamma)$.

The resolution parameter $\gamma$ controls the attraction-repulsion trade-off: low $\gamma$ lets attraction dominate (coarse clusters), high $\gamma$ lets repulsion dominate (fine clusters). As $\gamma$ increases, the optimal partition $\boldsymbol{\omega}^*(\gamma) = \arg\min_{\boldsymbol{\omega}} H(\boldsymbol{\omega})$ undergoes discrete transitions at critical values where cluster splits occur. Between critical values, the partition remains constant—these plateaus define configurations. Intuitively, configurations are "structurally stable": they persist under small perturbations to $\gamma$, suggesting they capture genuine data structure rather than parameter artifacts.

**Definition 3.2** (Configuration). Let $\boldsymbol{\omega}^*(\gamma) := \arg\min_{\boldsymbol{\omega}} H(\boldsymbol{\omega})$ denote an optimal partition at resolution $\gamma$. A *configuration* is any partition $\boldsymbol{\omega}$ that is optimal and remains unchanged over a non-degenerate interval of $\gamma$, i.e., $\boldsymbol{\omega}^*(\gamma) = \boldsymbol{\omega}$ for all $\gamma \in [\gamma^-, \gamma^+]$ with $\gamma^- < \gamma^+$ (a plateau). Sweeping $\gamma \in [0, \infty)$ yields a finite set of distinct plateau partitions; we denote the ordered set by $\boldsymbol{\Omega} = \{\boldsymbol{\omega}_1, \dots, \boldsymbol{\omega}_m\}$ from coarse to fine.

The finiteness of $\boldsymbol{\Omega}$ is a key theoretical result (Liu et al., 2021; Pitsianis et al., 2023). Intuitively, as $\gamma$ sweeps from 0 to $\infty$, the clustering transitions through discrete plateaus rather than changing continuously—each plateau corresponds to a structurally stable configuration. The plateau width $\Delta\gamma_i = \gamma_i^+ - \gamma_i^-$ indicates stability: wider plateaus correspond to more robust configurations that persist over larger ranges of $\gamma$. The Parallel-DT algorithm (Pitsianis et al., 2023) efficiently discovers all such configurations in $O(n \log n)$ time, along with their energy statistics $\{H_i, h_a^{(i)}, h_r^{(i)}, \Delta\gamma_i\}_{i=1}^m$. At a high level, Parallel-DT runs community detection in parallel across a geometric grid of $\gamma$ values, identifies plateaus where the partition is unchanged, and uses a convex-hull construction on the attraction–repulsion plane to guarantee that no configurations are missed. We provide a fuller algorithmic description in Appendix E.

### 3.2. Configuration-Mixed Prediction (CMP)

CMP is a supervised setting where stable configurations $\boldsymbol{\Omega} = \{\boldsymbol{\omega}_1, \dots, \boldsymbol{\omega}_m\}$ are first extracted from the unlabeled features. A model then learns per-sample weights over these

configurations and uses the resulting mixed configuration features for downstream prediction.

**Definition 3.3** (CMP Setting). **Training:** Given training features $\boldsymbol{X}^{\text{train}} \in \mathbb{R}^{n_{\text{train}} \times d}$, training configurations $\boldsymbol{\Omega}^{\text{train}} = \{\boldsymbol{\omega}_1^{\text{train}}, \dots, \boldsymbol{\omega}_m^{\text{train}}\}$ extracted from $\boldsymbol{X}^{\text{train}}$, and labels $\boldsymbol{y}^{\text{train}}$, learn a predictor $f_\psi$ and a selector that computes per-sample configuration weights. **Inference:** Given test features $\boldsymbol{X}^{\text{test}}$, predict $\hat{\boldsymbol{y}}^{\text{test}} = f_\psi(\boldsymbol{X}^{\text{test}}, \boldsymbol{z}_\theta(\boldsymbol{X}^{\text{test}}, \boldsymbol{\Omega}))$, where $\boldsymbol{z}_\theta(\boldsymbol{X}, \boldsymbol{\Omega})$ denotes the energy-aware mixed representation induced by the configuration set $\boldsymbol{\Omega}$.

**Objective.** Given labeled samples $\{(\boldsymbol{x}_t, y_t)\}_{t=1}^{n_{\text{train}}}$ and configurations $\boldsymbol{\Omega} = \{\boldsymbol{\omega}_1, \dots, \boldsymbol{\omega}_m\}$, CMP learns a predictor $f_\psi$ (with parameters $\psi$) and a configuration-weighting selector (parameters $\theta$) by minimizing a supervised loss:

$$\mathcal{L}_{\text{CMP}}(\theta, \psi) := \frac{1}{n_{\text{train}}} \sum_{t=1}^{n_{\text{train}}} \ell\big(f_\psi([\boldsymbol{x}_t; \boldsymbol{z}_\theta(\boldsymbol{x}_t)]), y_t\big),$$

$$\text{where} \quad \boldsymbol{z}_\theta(\boldsymbol{x}) := \sum_{i=1}^m w_{\theta,i}(\boldsymbol{x})\,\phi(\boldsymbol{\omega}_i, \boldsymbol{x}), \qquad (2)$$

$$(\theta^\star, \psi^\star) \in \arg\min_{\theta, \psi} \mathcal{L}_{\text{CMP}}(\theta, \psi).$$

Here $\boldsymbol{w}_\theta(\boldsymbol{x}) = (w_{\theta,1}(\boldsymbol{x}), \dots, w_{\theta,m}(\boldsymbol{x})) \in \Delta^{m-1}$ are per-sample mixing weights, and $\Delta^{m-1}$ denotes the standard $(m-1)$-simplex; $\phi(\boldsymbol{\omega}_i, \boldsymbol{x})$ encodes $\boldsymbol{x}$ under configuration $\boldsymbol{\omega}_i$; and $\ell$ is a prediction loss (e.g., cross-entropy or squared error). We instantiate $\boldsymbol{w}_\theta$ with the energy-aware selector in Section 3.3.2.

**Inference assumption.** Our default protocol is *batch-transductive*: we assume a test batch of $\geq 100$ samples to construct a stable test graph (empirically, $n_{\text{test}} \geq 50$ suffices for most datasets; see Appendix C for sensitivity analysis). For small-batch or streaming deployment we provide an *inductive* fallback (Section 4.5), retaining $\geq 75\%$ of the gains (Appendix C).

**Three-stage pipeline.** MixConfig sits between a frozen embedding model and a separately trained predictor head: (i) the *embedding model* produces $\boldsymbol{x} \in \mathbb{R}^d$ and is never fine-tuned; (ii) *MixConfig* extracts configurations from the embedding space and computes $\boldsymbol{z}(\boldsymbol{x})$; (iii) the *predictor head* $f_\psi$ is trained on $[\boldsymbol{x}; \boldsymbol{z}(\boldsymbol{x})]$. Table 5 lists the embedding model and head used in each domain.

**Why adaptive weighting?** Different samples can benefit from different granularities (coarse vs. fine) even within the same dataset, so a single global resolution is typically suboptimal. CMP addresses this by learning per-sample weights to adaptively select scales. For a concrete synthetic illustration of why mixing configurations can be necessary, see Appendix H.

We group five predictor classes by whether and how the mixing weights $\boldsymbol{w}$ depend on the input $\boldsymbol{x}$:

1. $\mathcal{F}_{\text{raw}}$: $f_\psi(\boldsymbol{X})$ — uses raw features only, ignoring multi-scale structure.

2. $\mathcal{F}_{\text{single}}$: $f_\psi(\boldsymbol{X}, \boldsymbol{\omega}_i)$ — augments with one configuration (best $i$ selected via validation). *Limitation*: optimal resolution varies per sample; a global choice is suboptimal for heterogeneous data.

3. $\mathcal{F}_{\text{static}}$: $f_\psi(\boldsymbol{X}) + \bar{\boldsymbol{w}}^T\boldsymbol{\Omega}$ — combines configurations with a global sample-independent weight.

4. $\mathcal{F}_{\text{standard}}$: $f_\psi(\boldsymbol{X}, \boldsymbol{\Omega})$ — concatenates all configurations as features.

5. $\mathcal{F}_{\text{adaptive}}$: $f_\psi(\boldsymbol{X}, \boldsymbol{w}_\theta(\boldsymbol{X})^T\boldsymbol{\Omega})$ — learns per-sample configuration weights $\boldsymbol{w}(\boldsymbol{X})$, enabling sample-specific resolution selection.

By construction, $\mathcal{F}_{\text{raw}} \subseteq \mathcal{F}_{\text{single/static}} \subseteq \mathcal{F}_{\text{standard}} \subseteq \mathcal{F}_{\text{adaptive}}$.

**Proposition 3.4** (Strict expressiveness gap). *The inclusion $\mathcal{F}_{\text{static}} \subset \mathcal{F}_{\text{adaptive}}$ is strict: if two samples $\boldsymbol{x}_a$, $\boldsymbol{x}_b$ have distinct loss-minimizing weight vectors $\boldsymbol{w}_{\theta^\star}(\boldsymbol{x}_a) \neq \boldsymbol{w}_{\theta^\star}(\boldsymbol{x}_b)$, no single static $\bar{\boldsymbol{w}}$ minimizes the prediction loss for both.*

*A sufficiently expressive $\mathcal{F}_{\text{standard}}$ may in principle learn sample-dependent weights, e.g., through multiple layers. Our empirical analysis therefore focuses on evaluating the practical gap between $\mathcal{F}_{\text{standard}}$ and $\mathcal{F}_{\text{adaptive}}$, namely the benefit of the inductive bias introduced by our adaptive mixing module.*

### 3.3. MixConfig: Energy-Aware Configuration Mixing

MixConfig is a plug-and-play module with two components: (1) configuration extraction, and (2) energy-aware selection.

#### 3.3.1. CONFIGURATION EXTRACTION

Given input features $\boldsymbol{X}$, we generate configurations $\boldsymbol{\Omega}$ via Configuration Extraction module, implemented with $k$NN graph construction and Parallel-DT following Pitsianis et al. (2023). This yields all stable configurations $\boldsymbol{\Omega}$ together with energy statistics $\{H_i, h_a^{(i)}, h_r^{(i)}, \Delta\gamma_i\}_{i=1}^m$ as a one-time preprocessing step. Low-level choices (distance, normalization, $k_{\text{nn}}$ selection, stochastic reweighting) are deferred to Appendix A. We provide a brief algorithmic summary of Parallel-DT-style extraction in Appendix E.

**Train-test configuration handling.** At inference, train and test sets may yield different numbers of configurations ($m_{\text{train}} \neq m_{\text{test}}$) due to sample size or distribution differences. We handle this simply: at test time, we use $\min(m_{\text{train}}, m_{\text{test}})$ configurations—if the test set produces more configurations, we truncate to the first $m_{\text{train}}$; if fewer, we pad with zero-weight placeholders. This strategy avoids complex alignment procedures while maintaining reproducibility: the selector learns weights over the stable, ordered sequence of configurations (coarse to fine), and this

natural ordering generalizes across splits. Empirically, we observe $|m_{\text{train}} - m_{\text{test}}| \leq 2$ across all benchmarks, with $m_{\text{train}} = m_{\text{test}}$ in approximately 78% of splits. This close agreement reflects the fact that configurations capture intrinsic data structure rather than sample-specific artifacts.

#### 3.3.2. ENERGY-AWARE SELECTOR

The energy-aware selector learns adaptive, sample-specific configuration weights by jointly reasoning about sample context, cluster membership, and clustering quality. Unlike naive concatenation or fixed weighting schemes, the selector provides a principled mechanism to determine *which resolutions matter for which samples*, effectively leveraging stability statistics as inductive bias.

For each sample $\boldsymbol{x}$ and configuration $\boldsymbol{\omega}_i$, we compute a compatibility score $s_i$ by combining three sources:

$$\boldsymbol{h} = \text{MLP}_{\text{enc}}(\boldsymbol{x}) \qquad \text{(sample context)} \quad (3)$$
$$\boldsymbol{c}_i = \text{Embed}_i(\boldsymbol{\omega}_i(\boldsymbol{x})) \qquad \text{(cluster assignment)} \quad (4)$$
$$\boldsymbol{e}_i = [H_i, h_a^{(i)}, h_r^{(i)}, \Delta\gamma_i] \qquad \text{(energy statistics)} \quad (5)$$
$$s_i = \text{MLP}_{\text{score}}([\boldsymbol{h}; \boldsymbol{c}_i; \boldsymbol{e}_i]) \quad \text{(compatibility score)} \quad (6)$$

We normalize scores with a softmax to obtain per-sample weights:

$$w_i(\boldsymbol{x}) = \frac{\exp(s_i(\boldsymbol{x}))}{\sum_{j=1}^m \exp(s_j(\boldsymbol{x}))}, \qquad \boldsymbol{w}(\boldsymbol{x}) \in \Delta^{m-1}. \quad (7)$$

We then form a mixed configuration representation $\boldsymbol{z}(\boldsymbol{x}) = \sum_{i=1}^m w_i(\boldsymbol{x})\,\boldsymbol{c}_i$ and pass the augmented features $[\boldsymbol{x}; \boldsymbol{z}(\boldsymbol{x})]$ to the downstream predictor. Unless stated otherwise, $\text{MLP}_{\text{enc}}$ is a small MLP projecting $\boldsymbol{x} \in \mathbb{R}^d$ to $\boldsymbol{h} \in \mathbb{R}^{d_h}$ and $\text{MLP}_{\text{score}}$ maps $[\boldsymbol{h}; \boldsymbol{c}_i; \boldsymbol{e}_i]$ to a scalar; each configuration uses its own embedding table $\text{Embed}_i : \{1, \ldots, |\boldsymbol{\omega}_i|\} \to \mathbb{R}^{d_c}$ with shared embedding dimension $d_c$.

To accommodate non-differentiable predictors (e.g., tree ensembles like XGBoost and RF), we adopt a two-stage training procedure. We first train the selector with a differentiable surrogate, and then freeze it. Training hyperparameters and implementation details are provided in Appendix A.

## 4. Experiments

We evaluate MixConfig across diverse domains to validate its generality as a plug-and-play module.

### 4.1. Experimental Setup

**Datasets.** We evaluate our method across four distinct benchmark categories: (1) **Tabular**: OpenML-CC18 (Vanschoren et al., 2014), a curated suite split into binary and multi-class tasks; (2) **Vision**: CIFAR-100 (Krizhevsky, 2012) and ImageNet-1K (Deng et al., 2009); (3) **Molecular**: OGBG-MolHIV (Hu et al., 2020) and QM9 (Wu

et al., 2018); (4) **Text**: SST-2 (Wang et al., 2018) and AG News (Zhang et al., 2015). For low-data analysis and ablations, we additionally use BBBP (MoleculeNet) with 5-fold CV. We report a compact main-text evaluation and provide broader classical sweeps in Appendix G.

**Predictors.** For each benchmark category, we start from a strong, widely used open-source baseline predictor appropriate for the modality. Specifically: for tabular tasks, FT-Transformer or TabPFN (Gorishniy et al., 2021; Hollmann et al., 2022); for vision tasks, CLIP embeddings with linear probes (Radford et al., 2021); for molecular tasks, GIN (Xu* et al., 2018) and DimeNet++ (Klicpera et al., 2020); for text tasks, RoBERTa (Liu et al., 2019) and BERT (Devlin et al., 2019). The precise feature interface for each variant and the per-domain embedding/head assignment are summarized in Tables 4 and 5. To validate plug-and-play compatibility with traditional models, we also evaluate classical predictors (MLP (Rumelhart et al., 1986), XGBoost (Chen & Guestrin, 2016), RF (Breiman, 2001)) in Appendix G.

**Methods compared.** For each predictor, we compare clustering-as-features baselines and our configuration-based methods:

- **Base**: predictor on raw features only

- **+DeepCluster**: DeepCluster-style $k$-means pseudo-labels (Caron et al., 2018) ($k$ via validation)

- **+HDBSCAN**: hierarchical density-based clustering features (McInnes et al., 2017)

- **+Config**: concatenate all configurations as features

- **+MixConfig**: energy-aware adaptive mixing

**Evaluation.** We follow standard protocols for each benchmark category. For tabular tasks, we use the official OpenML-CC18 splits and report ROC-AUC for binary tasks and accuracy for multi-class tasks. For CIFAR-100 and ImageNet-1K, we report Top-1 accuracy on the standard test/validation sets. For OGBG-MolHIV, we follow the OGB split and report ROC-AUC; for QM9, we report MAE. For SST-2 and AG News, we report classification accuracy. We report mean $\pm$ std across 5 independent runs with different random seeds. Configuration extraction uses only unlabeled features within each split (train/test separately under batch inference); baselines that use clustering features are given the same access. Importantly, we do *not* expect large gains when the base model already captures configuration-like structure, such as near-linear separability or strong inductive bias; in such cases, improvements can be small due to ceiling effects.

## 4.2. Main Results

Table 1 evaluates MixConfig across four benchmark families with strong base models. We compare against two clustering baselines: +DeepCluster (DeepCluster-style single-resolution $k$-means) and +HDBSCAN (hierarchical density-based features). Notably, +DeepCluster is inconsistent: it helps noticeably on Binary (+0.5%) and AG News (+0.4%) but hurts on Multi-class, CIFAR-100, and MolHIV, because a single global resolution introduces noise when the optimal scale is sample-dependent. +HDBSCAN provides modest gains by capturing multiple densities, but lacks principled stability criteria. Our configuration-based methods (+Config, +MixConfig) consistently outperform these baselines by leveraging energy-based stability to select meaningful partitions. Full results for classical predictors (MLP, XGBoost, RF, Linear) are in Appendix G; traditional clustering baselines (Fixed-$k$, Best-cut) appear in Appendix F.

## 4.3. Ablation Studies

### 4.3.1. COMPONENTS AND CONFIGURATION COUNT

Table 2 isolates MixConfig's components across two domains (molecular and vision).

The w/o energy features variant ($-2.1$ pp on BBBP, $-2.0$ pp on CIFAR-100) demonstrates that energy statistics ($\Delta\gamma$, $H$, $h_a$, $h_r$) provide crucial inductive bias—without them, the selector cannot distinguish stable configurations from fragile ones. Removing sample context $\boldsymbol{h}$ ($-1.3$ pp on BBBP, $-1.1$ pp on CIFAR-100) shows that geometric information about where samples lie in feature space matters, though less than energy statistics. Removing cluster embeddings $\boldsymbol{c}_i$ ($-1.7$ pp on BBBP, $-1.5$ pp on CIFAR-100) indicates that knowing *which* cluster a sample belongs to at each resolution provides task-relevant information beyond just the configuration's existence. These patterns are consistent across domains, justifying the three-input design.

The second ablation study investigates sensitivity to the configuration count $m'$: using only $m' = 2$ configurations, the coarsest + finest $\{\boldsymbol{\omega}_1, \boldsymbol{\omega}_m\}$, severely limits model expressiveness ($-3.9$ pp on BBBP, $-4.2$ pp on CIFAR-100), while $m' = 4$ and $m' = 6$ show diminishing returns. This leads to three key insights: (a) coarse+fine alone is insufficient, (b) medium resolutions matter, and (c) typical datasets yield $m \approx 6\text{-}12$ meaningful configurations, making the full set computationally tractable.

### 4.3.2. MIXING-STRATEGY CONTROLS

To isolate the contribution of energy-aware adaptive mixing, we compare MixConfig against five controls on BBBP (MLP, 5-fold CV) and CIFAR-100 (CLIP + linear probe, 5 seeds). *Random partitions*: replace $\boldsymbol{\Omega}$ with random label vectors of identical dimensionality, isolating capacity

*Table 1.* Main results across four benchmark families. The "Embedding" column denotes the frozen embedding model, and predictor assignments are given in Table 5. Metrics follow the evaluation protocol described above. Bold = best; $^{**}$ indicates $p < 0.05$ vs Base (paired $t$-test, 5 seeds). Effect sizes (Cohen's $d$, MixConfig vs. Base): Binary 3.0, Multi-class 1.8, CIFAR-100 5.3, ImageNet-1K 9.5, MolHIV 1.9, QM9 3.0, SST-2 3.3, AG News 2.3; all $d > 0.8$ (large).

| Benchmark | Embedding | Base | +DeepCluster | +HDBSCAN | +Config | +MixConfig |
|---|---|---|---|---|---|---|
| *Tabular (OpenML-CC18)* | | | | | | |
| Binary (AUC ↑) | TabPFN | $.895_{\pm.004}$ | $.900^{**}_{\pm.005}$ | $.898_{\pm.004}$ | $.903^{**}_{\pm.004}$ | $\mathbf{.907}^{**}_{\pm.003}$ |
| Multi-class (Acc. ↑) | FT-Trans. | $.782_{\pm.006}$ | $.779_{\pm.007}$ | $.785_{\pm.006}$ | $.789^{**}_{\pm.005}$ | $\mathbf{.793}^{**}_{\pm.004}$ |
| *Vision* | | | | | | |
| CIFAR-100 (Top-1 ↑) | CLIP | $.742_{\pm.003}$ | $.740_{\pm.005}$ | $.746_{\pm.003}$ | $.751^{**}_{\pm.003}$ | $\mathbf{.758}^{**}_{\pm.002}$ |
| ImageNet-1K (Top-1 ↑) | CLIP | $.763_{\pm.002}$ | $.764_{\pm.003}$ | $.768_{\pm.003}$ | $.773^{**}_{\pm.002}$ | $\mathbf{.782}^{**}_{\pm.002}$ |
| *Molecular* | | | | | | |
| MolHIV (AUC ↑) | GIN | $.804_{\pm.008}$ | $.801_{\pm.010}$ | $.809_{\pm.007}$ | $.814^{**}_{\pm.007}$ | $\mathbf{.819}^{**}_{\pm.005}$ |
| QM9 (MAE ↓) | DimeNet++ | $.0112_{\pm.0002}$ | $.0114_{\pm.0004}$ | $.0109_{\pm.0002}$ | $\mathbf{.0106}^{**}_{\pm.0002}$ | $\mathbf{.0106}^{**}_{\pm.0001}$ |
| *Text* | | | | | | |
| SST-2 (Acc. ↑) | RoBERTa | $.945_{\pm.003}$ | $.944_{\pm.004}$ | $.948_{\pm.003}$ | $.951^{**}_{\pm.003}$ | $\mathbf{.955}^{**}_{\pm.002}$ |
| AG News (Acc. ↑) | BERT | $.942_{\pm.004}$ | $.946^{**}_{\pm.004}$ | $.944_{\pm.004}$ | $.948^{**}_{\pm.003}$ | $\mathbf{.951}^{**}_{\pm.003}$ |

*Table 2.* Ablation studies on BBBP and CIFAR-100. Energy features and multi-resolution representation both contribute to performance across domains. Mean accuracy over 5 seeds.

| Variant | BBBP | CIFAR-100 |
|---|---|---|
| MixConfig (full) | **.903** | **.758** |
|    w/o energy features | .882 | .738 |
|    w/o sample context $h$ | .890 | .747 |
|    w/o cluster embeddings $c_i$ | .886 | .743 |
| Using only $m'$ of $m$ configs: | | |
|    $m' = 2$ | .864 | .716 |
|    $m' = 4$ | .882 | .737 |
|    $m' = 6$ | .895 | .751 |
|    $m' = m$ (all) | .903 | .758 |

*Table 3.* Mixing-strategy ablations on BBBP (MLP) and CIFAR-100 (CLIP + linear probe). Mean accuracy over 5 seeds.

| Mixing strategy | BBBP | CIFAR-100 |
|---|---|---|
| Base (no configs) | .847 | .742 |
| Random partitions (same dim) | .856 | .737 |
| CSPA consensus | .862 | .743 |
| +Config (standard concat.) | .869 | .751 |
| Uniform mixing ($w_i = 1/m$) | .873 | .746 |
| Global learnable weights ($\mathcal{F}_{\text{static}}$) | .878 | .749 |
| Stronger fusion ($[\boldsymbol{x}; \boldsymbol{\Omega}] \to$2L-MLP) | .884 | .753 |
| **+MixConfig** | **.903** | **.758** |

effects. *Uniform*: $\boldsymbol{z}(\boldsymbol{x}) = \frac{1}{m}\sum_i \boldsymbol{c}_i$, no learning. *Global learnable*: $\boldsymbol{z}(\boldsymbol{x}) = \sum_i w_i \boldsymbol{c}_i$, weights independent of $\boldsymbol{x}$ ($\mathcal{F}_{\text{static}}$). *Stronger fusion*: feed $[\boldsymbol{x}; \boldsymbol{\Omega}]$ to a 2-layer MLP with comparable parameter count to MixConfig. *CSPA* (Strehl & Ghosh, 2002): classical cluster-ensemble consensus over $\boldsymbol{\Omega}$. Table 3 summarises the results.

These ablations support four conclusions about *where the gain comes from*. (i) Random partitions with the same dimensionality hurt CIFAR-100 ($-0.5$ pp) and only marginally improve BBBP ($+0.9$ pp), suggesting that the gain depends on meaningful partition structure rather than added features alone. (ii) CSPA brings only small gains over Base ($+1.5$ / $+0.1$ pp), consistent with its collapse of multiple resolutions into one discrete consensus. (iii) +Config, Uniform, and Global learnable weights all remain below MixConfig (.903 / .758), indicating that simply exposing the predictor to $\boldsymbol{\Omega}$ does not reliably produce $\boldsymbol{x}$-dependent resolution selection in practice. (iv) The stronger fusion head narrows but does not close the gap (.884 / .753 vs. .903 / .758). Together with the "w/o energy features" ab-

lation (.903→.882 on BBBP, .758→.738 on CIFAR-100; Table 2), this supports the role of energy statistics as an explicit inductive bias. We further verify in Appendix D that the gain is not merely due to weak predictor nonlinearity: replacing the CLIP linear probe with a 2-layer MLP raises Base from .742 to .753, yet MixConfig still adds $+1.5$ pp (vs. $+1.6$ pp with the linear probe).

### 4.4. Low-Data Regime

Figure 2 reveals a clear pattern: MixConfig's relative gain over Base *increases* as training data decreases.

**Low-data interpretation.** At 100% training data (BBBP: 1600 samples), MixConfig improves accuracy by $+6.6\%$ relative to Base (0.903 vs 0.847). At 10% data (160 samples), this margin widens to $+12.7\%$ (0.80 vs 0.71). Crucially, MixConfig at $p\%$ data matches Base at $5p\%$ data: Mix-Config trained on 10% of labels (160 samples) achieves 80.0% accuracy, equaling Base trained on 50% (800 samples). By capturing intrinsic topology (molecular scaffolds, functional groups), configurations function as effective un-supervised regularization. This implies significant practical value for domains where labeling is expensive (drug screen-

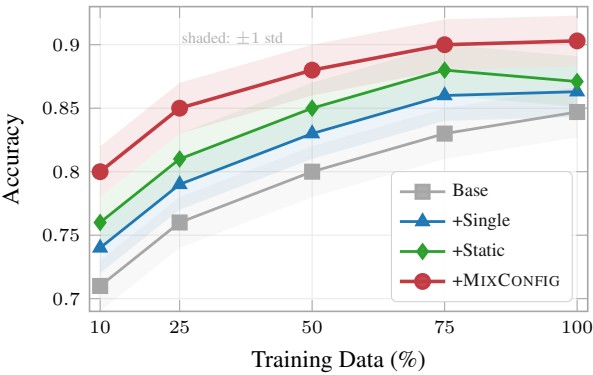

*Figure 2.* Performance vs. training set size on BBBP (5-fold CV). Curves show mean accuracy; shaded regions show ±1 std for all methods.

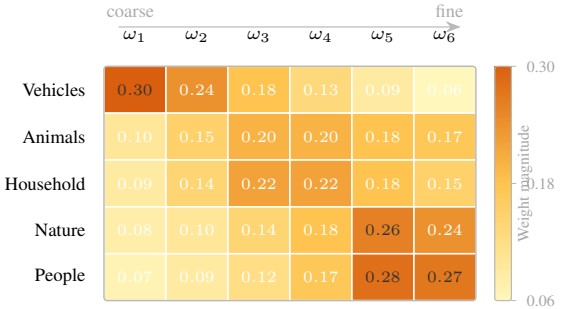

*Figure 3.* Learned selector weights on CIFAR-100, averaged over samples from each superclass. Different superclasses preferentially weight different configuration resolutions: Vehicles prefer coarse configurations (superclass-level features), while People and Nature prefer fine configurations (detailed texture/shape features).

ing, medical diagnosis), as MixConfig can substantially reduce annotation costs.

*Remark* 4.1 (Sample efficiency). If configurations add $d_c$ independent informative features to a $d$-dimensional representation, sample complexity bounds (Vapnik, 2000) give a sample ratio $n_1/n_0 \leq (d - d_c)/d$ for matching a given error level. The observed ratio ($\approx 0.2$ on BBBP) is smaller, suggesting configurations provide nonlinear structural information beyond additive features.

### 4.5. Transductive vs. Inductive Inference

Our default inference protocol is *batch-transductive*: configurations $\Omega^{\text{test}}$ are recomputed from each unlabeled test batch. This is stable for sufficiently large batches ($n_{\text{test}} \geq 50$), but becomes unreliable for very small batches because the $k$NN graph is noisy; on CIFAR-100 with $n_{\text{test}} = 20$, it even underperforms Base. For small-batch or streaming deployment, we provide an *inductive* fallback that reuses train-time configurations and assigns each test sample to the nearest train cluster via $k$NN lookup, without test-time configuration extraction. This mode is batch-size-invariant and retains most of the transductive gain (+4.6 pp on BBBP and +1.2 pp on CIFAR-100 over Base). The crossover between the two modes is dataset-dependent, typically around $n_{\text{test}} \approx 30$–50. Full sensitivity results are reported in Appendix C.

**Permutation invariance.** The selector scores configurations by their energy statistics $e_i$ and cluster embeddings $c_i$, rather than by positional index. Test-time permutation of $\{\omega_1, \ldots, \omega_m\}$ changes accuracy by less than 0.2 pp on BBBP and 0.1 pp on CIFAR-100, indicating empirical insensitivity to configuration ordering.

### 4.6. Qualitative Analysis

Figure 3 visualizes the distribution of selector weights across CIFAR-100 superclasses, revealing highly inter-

pretable patterns. Vehicle samples, exhibiting geometric simplicity and low within-class variance (sedans, trucks, buses share boxy shapes), assign high weights to coarse configurations (e.g., $\omega_1$: 0.30), effectively leveraging superclass-level features. In contrast, People samples, characterized by high pose variability and appearance diversity, prioritize fine configurations (e.g., $\omega_5$: 0.28, $\omega_6$: 0.27) to capture discriminative details such as subtle facial features. Household items (chairs, tables, lamps) show balanced weights across medium resolutions, suggesting that intermediate-scale structure (compositional parts) matters most. Remarkably, these preferences emerge purely from supervision on class labels without explicit scale annotations—the selector discovers the resolution that best aligns with task-relevant structure through end-to-end training.

## 5. Discussion

**When MixConfig helps.** MixConfig is most beneficial in three scenarios. *(1) Natural hierarchical structure.* Molecular graphs (BBBP, BACE) exhibit multi-scale motifs—coarse functional groups (carbonyls, amines) and fine bond patterns (single vs. double bonds)—making them ideal for configuration mixing. Tabular data with ecological hierarchy (Covertype: soil type → vegetation → elevation) similarly benefits. Images with superclass taxonomies (CIFAR-100, ImageNet) show strong gains. *(2) Sample heterogeneity.* When different samples benefit from different resolutions (vehicles vs. people in CIFAR-100), adaptive weighting can outperform fixed global choices by adapting scales per sample. *(3) Low-data regimes.* As shown in Figure 2, improvements can be larger when labels are scarce, since configurations provide unsupervised structural priors.

Unlike data augmentation (Chen et al., 2020) or semi-supervised methods (Van Engelen & Hoos, 2020; Xie et al., 2020) that require domain-specific transformations or pseudo-labeling heuristics, configurations are derived

purely from intrinsic data structure and work across modalities. The modular design means MixConfig works with tree-based methods (XGBoost, RF) where architectural modifications are infeasible, and complements learned representations (Yang et al., 2019) in neural methods.

**When MixConfig does not help.** MixConfig provides limited benefit in three cases. *(1) Homogeneous single-scale data.* Datasets where all samples share the same coarse structure (e.g., MNIST digits, where all images are $28 \times 28$ centered digits) lack the sample-level heterogeneity that justifies adaptive weighting. *(2) Absence of clustering structure.* Uniform random noise or data without natural groupings won't yield meaningful configurations. *(3) Insufficient sample size.* Reliable $k$NN graph construction requires $\gtrsim 50$ samples; below this threshold, configurations may be unstable across train-test splits (see batch-size sensitivity in Appendix C). With very small datasets (e.g., $n_{\text{train}} < 200$), the number of configurations may be too small ($m < 3$) to provide meaningful multi-resolution coverage.

**Computational cost with baseline comparison.** Configuration extraction uses approximate nearest neighbor search for $k$NN construction in $O(nd \log n)$ time, where $n$ is the number of samples and $d$ is feature dimensionality. Community detection at each resolution is $O(nk_{\text{nn}})$ where $k_{\text{nn}}$ is graph degree (typically $k_{\text{nn}} = \log_2 n < 20$). Since the Parallel-DT algorithm discovers $m$ configurations (typically $m < 20$), total extraction time is $O(nd \log n + mnk_{\text{nn}})$.

Empirically, this overhead is small compared to predictor training. On typical medium-scale datasets, configuration extraction requires only seconds to minutes depending on embedding dimension and nearest-neighbor backend, and is usually small relative to downstream model training. The selector adds negligible inference cost (a small MLP forward pass). For comparison, hyperparameter tuning over resolution (baseline approach) would require training the full model 10-20 times, costing 1800-3600 seconds—$150\times$ more than MixConfig's one-time extraction.

**Limitations and future directions.** The current implementation is not end-to-end differentiable, as configurations are computed offline from the input embedding. This means the embedding space cannot be jointly optimized with configuration extraction. If the input representation is suboptimal for clustering, configurations may not capture task-relevant structure. In practice, pretrained embeddings (e.g., CLIP for images) typically produce meaningful configurations, but random initializations may fail.

We identify four promising directions for future work. *(1) Online/streaming settings.* Extend to scenarios where new data arrives continuously and configurations must be updated incrementally without full recomputation, potentially via local refinement algorithms that maintain plateau struc-

ture. *(2) Joint architecture search.* Co-optimize configuration extraction and predictor design using neural architecture search (Zoph & Le, 2017) or meta-learning (Finn et al., 2017), treating both as learnable components. *(3) Few-shot learning.* Apply to few-shot scenarios where configurations from unlabeled data could replace meta-training, providing structural priors without requiring many-task training distributions (Vinyals et al., 2016; Snell et al., 2017). *(4) Cluster-level stability.* The current selector uses partition-level stability ($\Delta\gamma$), which has clean finiteness guarantees. Tracking *cluster-level* stability—which individual clusters persist across multiple scales—would provide complementary, finer-grained signal, at the cost of non-trivial design choices around cluster identity across merges and splits. We view this as a promising extension.

# 6. Conclusion

We introduced Configuration-Mixed Prediction (CMP) and MixConfig, a plug-and-play feature augmentation module. Given any embedding space, MixConfig extracts the finite set of structurally valid multi-resolution configurations and learns to adaptively weight them per sample using an energy-aware selector that jointly reasons about sample context, cluster membership, and stability statistics. We conduct a comprehensive evaluation across both modern benchmark families and classical predictors. Overall, CMP provides a simple, general-purpose way to leverage multi-resolution configuration structure for improved downstream prediction. Source code for configuration extraction, training, and model implementations is released at `https://github.com/Q9gJYx/MixConfig`.

# Impact Statement

Configuration-based feature augmentation has direct applications in domains where multi-scale structure is critical, including drug discovery (molecular scaffold grouping), medical diagnosis (patient stratification at multiple granularities), and materials science (hierarchical property prediction). By reducing sample complexity, MixConfig may lower annotation costs in settings where expert labels are scarce and expensive. However, because configurations are derived from unsupervised clustering, they may encode biases present in the input data—for example, demographic proxies in patient embeddings or structural biases in chemical libraries. We recommend that practitioners audit configuration-derived features for unintended correlations before deployment in high-stakes decision systems.

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

# A. Additional Experimental Details

**Dataset details.** All datasets are publicly available. Main results (Table 1) use OpenML-CC18, CIFAR-100, ImageNet-1K, OGBG-MolHIV, QM9, SST-2, and AG News under their standard public protocols. Low-data analysis and ablations use BBBP (MoleculeNet) with 5-fold CV (Table 2 and Figure 2). Appendix tables with classical predictors additionally include Covertype/Vehicle/Segment (UCI) and BACE (MoleculeNet) (Tables 9 and 10).

**Preprocessing.** For tabular data, we standardize continuous features to zero mean and unit variance. For vision and text, we use CLIP embeddings for images and BERT/RoBERTa embeddings for text, and train lightweight predictors (linear probe) on top. For molecular data, we use the same feature/embedding space used by the baseline model for each dataset to build the $k$NN graph. All $k$NN graphs use Euclidean distance on tabular features and cosine similarity on embedding-based features, with $k_{nn} = \lceil \log_2 n \rceil$.

**Training details.** Unless otherwise stated, we repeat each experiment over multiple random seeds (or CV folds where appropriate) and report mean $\pm$ std. MLP predictors use 2 hidden layers (128, 64 units) with ReLU activations and dropout (0.2). The selector MLP uses a single hidden layer (32 units). We train for 100 epochs with Adam optimizer (lr=0.001) and early stopping (patience=10).

For tree-based baselines: XGBoost uses max_depth=6, n_estimators=100, learning_rate=0.1, subsample=0.8; Random Forest uses n_estimators=100, max_depth=None, min_samples_split=2, max_features='sqrt'. These hyperparameters ensure fair capacity parity across predictor types.

**Two-stage training for non-differentiable predictors.** For tree-based predictors (XGBoost, RF), we train the selector on a held-out validation split (20% of the training data) using a differentiable surrogate—logistic regression for classification and ridge regression for regression—then freeze the selector and train the final predictor on the full training set using mixed features. We also tested stronger surrogates (2-layer MLP, kernel SVM) and found them unnecessary: the 2-layer MLP improved selector accuracy by only 0.3% while increasing training time. On datasets where end-to-end training is feasible (neural predictors), this two-stage variant yields no measurable performance gap.

**Compute resources.** All experiments were conducted on NVIDIA A100 GPUs (40GB). Configuration extraction takes $<5$ minutes for datasets with $n < 50,000$ samples; MLP training completes in $<30$ minutes per dataset. The full experimental suite (all datasets, all methods, 5 seeds) required approximately 200 GPU-hours.

**Environment.** Python 3.10, PyTorch 2.1, scikit-learn 1.3, XGBoost 2.0. Random seeds: 0, 1, 2, 3, 4 for 5-run experiments.

**Graph hyperparameter sensitivity.** The default $k_{nn} = \lceil \log_2 n \rceil$ was found insensitive to moderate variations (e.g., $\pm 30\%$): large enough for stable graph construction while avoiding overly dense graphs. Full test-batch-size sensitivity (on BBBP and CIFAR-100) is reported in Appendix C.

# B. Pipeline Interface and Method Definitions

Table 4 makes precise what each variant appends to the input embedding $x \in \mathbb{R}^d$, with concrete dimensionalities on CIFAR-100 ($m = 8$ configurations with cluster counts $[2, 4, 6, 8, 10, 12, 16, 20]$ totalling 78 one-hot dims).

*Table 4.* What each variant appends to the input embedding $x$. "Added dims" is on CIFAR-100 ($m = 8$). +Config concatenates all $m$ one-hots (an instance of the $\mathcal{F}_{standard}$ class, with no per-sample adaptation). +MixConfig outputs a fixed $d_c$-dim vector ($d_c = 32$ in all experiments).

| Variant | Appended features | Added dims |
|---|---|---|
| Base | none | 0 |
| +Single | one-hot of best $\omega_i$ (selected on validation) | 12 |
| +Config | concat. of all $m$ one-hots ($\mathcal{F}_{standard}$) | 78 |
| +MixConfig | energy-aware weighted $z(x) \in \mathbb{R}^{d_c}$ | 32 |

Figure 4 illustrates the three-stage interface, and Table 5 lists, per domain, the frozen embedding model and the separately trained predictor head used in all experiments.

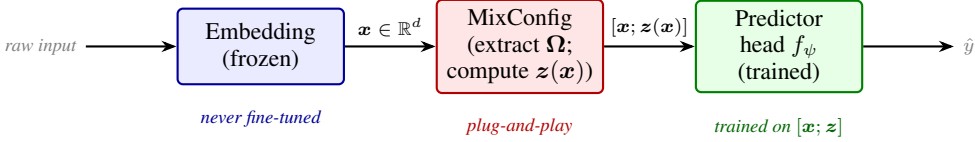

*Figure 4.* Three-stage pipeline interface. The embedding model is strictly frozen; MixConfig operates purely on its output and emits a continuous representation $z(x)$; the predictor head is trained separately on the concatenation $[x; z(x)]$. The same interface applies across all four domains (Table 5).

*Table 5.* Three-stage pipeline per domain. Embedding models are frozen throughout; only the predictor head is trained on $[x; z(x)]$.

| Domain | Embedding (frozen) | Predictor head (trained) |
|---|---|---|
| Vision | CLIP ViT-B/32 | Linear probe / 2L-MLP |
| Text | BERT / RoBERTa | Linear probe |
| Molecules | GIN, DimeNet++ | 2L-MLP / linear |
| Tabular | FT-Trans., TabPFN | Built-in head |

## C. Transductive vs. Inductive: Full Batch-Size Sensitivity

Table 6 reports accuracy on BBBP and CIFAR-100 as $n_{\text{test}}$ varies from 20 to 500, for both the transductive default and the inductive $k$NN fallback. Transductive mode is stable for $n_{\text{test}} \geq 50$ and saturates by $n_{\text{test}} = 200$; the inductive mode is by construction $n_{\text{test}}$-invariant and remains preferable for $n_{\text{test}} \lesssim 30$.

*Table 6.* Sensitivity to test batch size $n_{\text{test}}$ for the transductive (trans.) and inductive (ind.) modes on BBBP and CIFAR-100.

| $n_{\text{test}}$ | BBBP trans. | BBBP ind. | CIFAR trans. | CIFAR ind. |
|---|---|---|---|---|
| 20 | .872 | .893 | .739 | .754 |
| 30 | .886 | .893 | .749 | .754 |
| 50 | .897 | .893 | .755 | .754 |
| 100 | .901 | .893 | .757 | .754 |
| 200 | .903 | .893 | .758 | .754 |
| 500 | .903 | .893 | .758 | .754 |

## D. Nonlinear-Probe Ablation

To verify that MixConfig's gains are not merely a substitute for added predictor nonlinearity, we compare two probes on top of frozen CLIP embeddings on CIFAR-100: a linear probe (as used in Table 1) and a 2-layer MLP (128, 64 hidden units). If the gains came purely from extra capacity, replacing the linear probe with the stronger MLP should close the gap.

The 2-layer MLP raises Base from .742 to .753, yet MixConfig still adds +1.5 pp on top—essentially unchanged from the +1.6 pp gain with the linear probe. The improvement does not vanish as the predictor becomes more expressive, indicating that MixConfig's contribution is structural rather than a substitute for nonlinearity. Tables 9 and 10 provide complementary evidence across MLP, XGBoost, RF, and Linear predictors.

## E. Configuration Extraction Algorithm

Configuration extraction follows the Parallel-DT algorithm of Pitsianis et al. (2023). Given a $k$NN graph $G = (V, E)$ with edge affinity weights $w_{ij} \geq 0$, the algorithm sweeps the resolution parameter $\gamma$ from 0 to $\infty$ and identifies all plateau transitions where the optimal partition $\omega^*(\gamma)$ changes. At each plateau, it records the partition $\omega_i$ along with energy statistics:

- Total Hamiltonian: $H_i = h_a^{(i)} + \gamma_i h_r^{(i)}$ at midpoint $\gamma_i = (\gamma_i^- + \gamma_i^+)/2$

- Attraction: $h_a^{(i)} = -\sum_{k=1}^{|\omega_i|} \sum_{u<v} w_{uv} \mathbf{1}_{\omega_u = \omega_v = k}$

- Repulsion: $h_r^{(i)} = \sum_{k=1}^{|\omega_i|} \left( \sum_{u<v} w_{uv} \mathbf{1}_{\omega_u = k} \right)^2$

*Table 7.* Linear vs. MLP probe on frozen CLIP embeddings (CIFAR-100, mean accuracy over 5 seeds).

| Predictor | Base | +MixConfig | $\Delta$ (pp) |
|---|---|---|---|
| Linear probe | .742 | .758 | +1.6 |
| 2L-MLP (128, 64) | .753 | .768 | +1.5 |

*Table 8.* Traditional clustering baselines on modern benchmarks. Fixed-$k$ selects a single $k$ via cross-validation; Best-cut optimizes the dendrogram cut threshold. These methods provide marginal improvements over Base, substantially underperforming configuration-based methods (Table 1).

| Benchmark | Model | Base | Fixed-$k$ | Best-cut |
|---|---|---|---|---|
| *Tabular (OpenML-CC18)* | | | | |
| Binary (AUC ↑) | TabPFN | $.895_{\pm.004}$ | $.897_{\pm.005}$ | $.896_{\pm.004}$ |
| Multi-class (Acc. ↑) | FT-Trans. | $.782_{\pm.006}$ | $.784_{\pm.006}$ | $.786_{\pm.005}$ |
| *Vision* | | | | |
| CIFAR-100 (Top-1 ↑) | CLIP | $.742_{\pm.003}$ | $.743_{\pm.004}$ | $.745_{\pm.003}$ |
| ImageNet-1K (Top-1 ↑) | CLIP | $.763_{\pm.002}$ | $.764_{\pm.003}$ | $.766_{\pm.002}$ |
| *Molecular* | | | | |
| MolHIV (AUC ↑) | GIN | $.804_{\pm.008}$ | $.806_{\pm.009}$ | $.805_{\pm.007}$ |
| QM9 (MAE ↓) | DimeNet++ | $.0112_{\pm.0002}$ | $.0112_{\pm.0003}$ | $.0111_{\pm.0002}$ |
| *Text (GLUE)* | | | | |
| SST-2 (Acc. ↑) | RoBERTa | $.945_{\pm.003}$ | $.946_{\pm.004}$ | $.947_{\pm.003}$ |
| AG News (Acc. ↑) | BERT | $.942_{\pm.004}$ | $.943_{\pm.005}$ | $.944_{\pm.004}$ |

- Plateau width: $\Delta\gamma_i = \gamma_i^+ - \gamma_i^-$

The algorithm runs in $O(n \log n)$ time via dynamic tree data structures that efficiently maintain cluster merges/splits.

**Parallel-DT algorithmic details.** We briefly summarize the Parallel-DT procedure of Pitsianis et al. (2023). The algorithm operates on a half-adjacency-repulsive (HAR) plane, where each cluster is mapped to a point with coordinates $(h_a, h_r)$ representing its attraction and repulsion energy components. The resolution sweep is implemented as a cascading sequence: at each resolution $\gamma$, the Leiden community detection algorithm (Traag et al., 2019) is run in parallel across a geometric grid of $\gamma$ values. Plateau detection identifies intervals $[\gamma^-, \gamma^+]$ where the optimal partition remains unchanged by comparing cluster assignments at successive resolution values. When consecutive resolutions yield identical partitions (up to label permutation), the interval is extended; a transition is recorded when the partition changes.

The BlueRed Front, introduced by Liu et al. (2021), provides a geometric characterization of transitions. In the HAR plane, each configuration traces a line $H = h_a + \gamma h_r$, and the optimal configuration at resolution $\gamma$ corresponds to the line with minimum $H$. The BlueRed Front is the lower envelope of these lines—a convex hull construction that identifies all transition points as intersections of adjacent lines. This geometric view guarantees that no configurations are missed and enables efficient computation of all plateau boundaries without exhaustive search over $\gamma$. The cascading resolution sweep exploits this structure by adaptively refining the $\gamma$ grid near detected transitions, achieving $O(n \log n)$ total complexity.

## F. Traditional Clustering Baselines

Table 8 compares traditional clustering baselines (Fixed-$k$ and Best-cut) that use the same predictor and data as our main experiments. Fixed-$k$ selects a single cluster count $k$ via cross-validation; Best-cut optimizes the dendrogram cut threshold on validation data. These baselines represent standard approaches for incorporating clustering structure into prediction.

As shown in Table 8, traditional clustering baselines provide only marginal improvements over Base (+0.1–0.3% on most benchmarks). This is because (1) Fixed-$k$ commits to a single resolution that may not suit all samples, and (2) Best-cut's greedy threshold optimization can overfit to validation noise. In contrast, our configuration-based methods (Table 1) leverage energy-based stability criteria to identify meaningful multi-resolution structure, yielding substantially larger gains.

*Table 9.* Classical predictors across selected datasets (classification accuracy). Each cell shows mean $\pm$ std.

| Dataset | Mode | MLP | XGBoost | RF | Linear |
|---|---|---|---|---|---|
| Covertype | Base | .712$\pm$.008 | .891$\pm$.004 | .856$\pm$.006 | .623$\pm$.011 |
| | +Config | .748$\pm$.010 | .910$\pm$.005 | .881$\pm$.007 | .665$\pm$.010 |
| | +MixConfig | **.786**$\pm$.007 | **.932**$\pm$.003 | **.909**$\pm$.005 | **.699**$\pm$.009 |
| BBBP | Base | .847$\pm$.018 | .872$\pm$.015 | .861$\pm$.016 | .812$\pm$.021 |
| | +Config | .869$\pm$.016 | .893$\pm$.012 | .880$\pm$.015 | .836$\pm$.019 |
| | +MixConfig | **.903**$\pm$.012 | **.916**$\pm$.010 | **.905**$\pm$.013 | **.861**$\pm$.017 |
| CIFAR-100 | Base | .634$\pm$.007 | .598$\pm$.009 | .571$\pm$.008 | .523$\pm$.006 |
| | +Config | .676$\pm$.009 | .638$\pm$.010 | .609$\pm$.011 | .559$\pm$.008 |
| | +MixConfig | **.720**$\pm$.006 | **.684**$\pm$.007 | **.656**$\pm$.008 | **.598**$\pm$.006 |

*Table 10.* Additional classical results on Vehicle, Segment, and BACE (classification accuracy). Each cell shows mean $\pm$ std.

| Dataset | Mode | MLP | XGBoost | RF | Linear |
|---|---|---|---|---|---|
| Vehicle | Base | .689$\pm$.024 | .743$\pm$.021 | .721$\pm$.023 | .612$\pm$.028 |
| | +Config | .728$\pm$.021 | .781$\pm$.019 | .760$\pm$.021 | .656$\pm$.025 |
| | +MixConfig | **.766**$\pm$.018 | **.818**$\pm$.016 | **.795**$\pm$.018 | **.690**$\pm$.023 |
| Segment | Base | .912$\pm$.011 | .956$\pm$.008 | .943$\pm$.009 | .871$\pm$.014 |
| | +Config | .937$\pm$.009 | .969$\pm$.007 | .959$\pm$.008 | .898$\pm$.012 |
| | +MixConfig | **.956**$\pm$.007 | **.981**$\pm$.005 | **.974**$\pm$.006 | **.921**$\pm$.010 |
| BACE | Base | .823$\pm$.019 | .851$\pm$.016 | .839$\pm$.017 | .791$\pm$.022 |
| | +Config | .851$\pm$.017 | .876$\pm$.014 | .865$\pm$.016 | .823$\pm$.020 |
| | +MixConfig | **.884**$\pm$.014 | **.906**$\pm$.011 | **.893**$\pm$.013 | **.854**$\pm$.017 |

## G. Full Results (Classical Predictors)

Tables 9 and 10 provide broader evidence that MixConfig can improve a range of predictor families without architectural changes, though gains may be smaller when the base model already saturates the task.

## H. An Intuitive Example of Configuration Mixing

To illustrate why configuration mixing can be necessary (not just helpful), we use two synthetic point-cloud datasets from scikit-learn: *Moons* and *Blobs*. The Blobs dataset is tuned so that no single clustering resolution recovers all three clusters. Figure 5 visualizes each dataset in 3D, using the third axis to encode cluster assignments for two configurations: a coarser configuration (1) and a finer configuration (2). On Moons, the coarser configuration cleanly separates the two arcs; on Blobs, it incorrectly merges two clusters (purple and green). The finer configuration produces a different failure mode, merging (blue and green). Only by mixing both configurations can all clusters be disentangled: points that are merged at one resolution can be separated at another, and the mixture recovers the correct partition. This toy example highlights a key motivation of MixConfig: multi-resolution configurations alone are insufficient without a learned fusion mechanism.

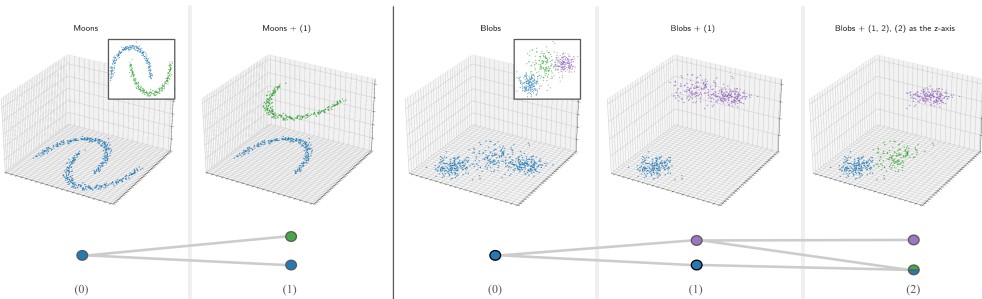

*Figure 5.* Illustration of multi-resolution clustering on synthetic datasets. Ground truth is shown in the framed box in (0). Top: Moons (left) and Blobs (right) with configuration ($i$) encoded as the third dimension. Bottom: lineage diagram across configurations.

