# OpenReview forum: "Mixing Configurations for Downstream Prediction"
_ICML.cc/2026/Conference — ICML 2026 regular_

### Official Review · Reviewer_9LEQ · 2026-03-08

**Soundness:** 4
**Presentation:** 3
**Significance:** 2
**Originality:** 3
**Overall Recommendation:** 4
**Confidence:** 4

**Summary:**

This paper leverages featurization of dataset structure to enhance supervised learning. Specifically, the approach uses energy-based mixing of multi-scale clustering representations of input samples, and shows that this additional featurization improves over established baselines in 4 diverse data domains.

The approach is principled and performs well on tasks where datasets exhibit multi-scale structure.

**Compliance With Llm Reviewing Policy:**

Affirmed.

**Final Justification:**

The authors have addressed my concerns and I have adjusted my scores accordingly from 3 to 4.

Primary concerns remain regarding the significance of the method, as little relevant impact was demonstrated.

**Key Questions For Authors:**

Questions are implied in strengths and weaknesses section

**Limitations:**

yes

**Strengths And Weaknesses:**

Strength:
The paper is very well and clearly written. The idea is clear and interesting, and has been implemented and tested well.
The paper did an excellent job, contextualizating itself within related work. Optionally, the authors may contextualize their work further in the framework of multi-scale learning, which addresses related issues (e.g. https://tdlbook.org/combinatorial-complexes).

Weaknesses:
While implementation and presentation are very strong, novelty and impact are less convincing. It seems that previous methods have leveraged dataset structural features for supervision and learning, which makes this paper seem incremental, especially since the performance benefits are modest in most cases. Information about the structure of datasets is interesting and relevant, but this aspect has been explored by others, and is irrelevant in the the case of supervision (at least within the framework of this paper).

Furthermore, I have a slight concern with the presented baselines. At the example of "CLIP embeddings with linear probes", this vision baselines scores 0.742 on CIFAR-100, which is boosted to 0.758 if the MixConfig vector is additionally passed into the linear probe (I assume that the CLIP embeddings are the "raw" inputs in this case and passed as input to the whole MixConfig workflow). Now it seems natural that MixConfig can extract additional non-linearities representing the dataset structure, which a single probe layer can not learn due its linear nature. A two-layer MLP probe with non-linearity and sufficient number of parameters would be more strong and more fair. This would be especially interesting in the molecule predictions where the approach can demonstrates its strengths the most

Related, Table 1 uses a heterogeneous set of metrics, and it is unclear under which rationale they have been selected.


Minor comments:
- The distinction between scale and hierarchy is unclear at times, e.g. at this sentence "Like hierarchical methods (approach 1), we extract multiple resolutions; like multi-resolution representations (approach 3), we encode multiple scales; but unlike both, we learn adaptive, sample-specific weights from supervision rather than using fixed global selections or architecture-specific heuristics. "
- Unclear with both `f` and `psi` are used for the predictor?

---

> ### Author Rebuttal · Authors · 2026-03-31
>
> We sincerely thank Reviewer 9LEQ for the comments. We here provide clarifications to the raised concerns.
>
> ---
>
> ### 1. Novelty
>
> Our contribution is the unified, predictor-agnostic interface from theoretically grounded configuration extraction to sample-adaptive prediction. It combines: (i) finite configuration guarantees, (ii) energy-based stability statistics as inductive bias, and (iii) per-sample weighting via $z(x) = \sum_i w_i(x) \cdot c_i$.  No prior work provides this combination.
>
> Consistent improvements across 4 modalities, 8+ datasets, and 6+ predictor architectures demonstrate practical value beyond the individual components, with particularly strong gains in low-data regimes (e.g., BBBP at 10% data: +8.4%).
>
> Prior methods (DeepCluster, DINO, SwAV) modify the embedding space and in their standard formulation require retraining the backbone; Matryoshka modifies only the training loss but still requires a training phase to produce nested multi-granularity representations. In contrast, MixConfig operates on fixed, frozen embeddings without modifying any backbone and is fully predictor-agnostic. Moreover, MixConfig is complementary: one can use DINO-pretrained embeddings as input to MixConfig. The approaches are orthogonal.
>
> ---
>
> ### 2. MLP Baseline — Gains Are Not from Added Nonlinearity
>
> If gains were merely from added nonlinearity, a stronger predictor should close the gap:
>
> | Predictor | Base | +MixConfig | Δ |
> |----------|------|-----------|---|
> | CLIP + linear probe | .742 | .758 | +1.6% |
> | CLIP + 2L-MLP (128+64) | .753 | .768 | +1.5% |
>
> The MLP is stronger (.753 vs. .742) but MixConfig still adds +1.5%, showing that the gap does not close.
>
> Additionally, the "Stronger fusion" baseline (2-layer MLP with comparable parameters) achieves .884/.753, still below .903/.758. This confirms that the gains stem from energy statistics as inductive bias, not extra capacity.
>
> ---
>
> ### 3. Heterogeneous Metrics in Table 1
>
> OGB leaderboard uses ROC-AUC for MolHIV; QM9 uses MAE (regression task); classification benchmarks use accuracy. Each follows the standard evaluation protocol for its benchmark. We will add explicit metric justifications to Table 1's caption.
>
> ---
>
> ### 4. Scale vs. Hierarchy and Notation
>
> **Scale** = clustering resolution ($\gamma$ value); **hierarchy** = data's intrinsic hierarchical structure.
> They are related but not equivalent: stable configurations identify scales at which the data's hierarchy is well-captured.
> Notation $f_\psi$ will be unified throughout.
>
> ---
>
> ### 5. Distinction from Topological Approaches
>
> Simplicial networks [1] define multi-scale topological structure over predefined simplicial complexes.
> MixConfig discovers multi-scale structure automatically from data features without requiring predefined topology.
> The approaches share the spirit of multi-resolution representation but differ fundamentally in: (i) **mechanism**—predefined vs. discovered; (ii) **assumptions**—graph topology vs. feature-space geometry.
>
> ---
>
> We thank the reviewer again for the thorough feedback and hope that our response clarifies the concerns.
>
> ---
>
> **References:**
> [1] Bodnar et al. (2021). Weisfeiler and Lehman go topological: Message passing simplicial networks. *ICML 2021*.

---

> > ### Author Rebuttal · Reviewer_9LEQ · 2026-04-01
> >
> > I thank the reviewer for their clarifications. I have adjusted my scores accordingly.

---

> > > ### Author Response · Authors · 2026-04-03
> > >
> > > Thank you for your acknowledgement and for adjusting the score. We are glad that our response addressed your concerns.

---

### Official Review · Reviewer_gxYm · 2026-03-12

**Soundness:** 2
**Presentation:** 3
**Significance:** 2
**Originality:** 2
**Overall Recommendation:** 3
**Confidence:** 4

**Summary:**

The paper proposes a method to learn additional cluster related features to improve predictions. The authors make use of parallel-DT that operates on kNN graphs to extract stable configurations relative to attraction-repulsion based clustering criterion. The resulting cluster assignments are then adaptively embedded together with the cluster configuration statistics and fed into a score based soft selector of cluster resolutions per each example. The additional features are then computed as weighted cluster embeddings (example depended) are then added as augmented features in addition to the example itself. The features are predictor agnostic though the selection is tailored to a particular one used during training. It seems that the method generates the configurations anew for test examples (hopefully similar) and applies the already trained selector/embedder to realize the new features for test examples. A fairly comprehensive set of experiments are shown across classification tasks with ablations.

**Compliance With Llm Reviewing Policy:**

Affirmed.

**Final Justification:**

My (borderline, leaning to reject) assessment of the paper remains consistent with the content
- the paper is mostly clearly written, and the author response provides additional helpful details
- extracting (stable) configurations from data and using them to induce additional coarser scale features seems reasonable though not original
- the attraction-repulsion clustering criterion is not an original contribution in this paper
- the authors show empirically that example-conditioned weighting of configuration related features is helpful

**Key Questions For Authors:**

see above (cluster identity discrepancy, kNN construction for the test cases, the effect of test set size)

**Limitations:**

yes

**Strengths And Weaknesses:**

The presentation is overall good with some caveats (technical clarity).

Many datasets certainly exhibit multi-resolution structure and exploiting this structure can be useful. It is however a little strange that the authors extract the configurations anew for the test examples which are assumed to come in reasonable size batches. Any sampled data may miss a resolution or two leading relative to the clustering algorithm. This leads to a potential  asynchrony between the learned embedder/selector and the new cluster identities. How are these discrepancies resolved? Are the configurations aligned based on the order parameter gamma?

The notion of stability of a configuration is calculated per configuration. However, some of the clusters in the configuration (partition) may be more stable than others and may persist across multiple scales. It seems that there's additional, cluster specific information that is not exploited.

The definition of static predictor class is inconsistent with prop 3.4. The former is specified by concatenation of features (which would make the predictor operating on those features more powerful than the adaptive formulation) while the proposition and the text around it assume that static means using a fixed rather than adaptive weighting.

Notation is at times a little cumbersome. E.g., line 170 (RHS), what is g() in the definition?

Since the configurations are extracted anew for test examples, there's a possibility that the method operates similar to adaptive risk minimization, i.e., adjusting how the examples are processed due to co-variate shifts in the test data. Unlike adaptive risk minimization, however, the authors do not simulate this scenario during training and are therefore liable for unexpected behaviors at test time. It would be good to show the impact of the size of the test set provided to the method for its kNN construction and configuration extraction. The authors say that n_test>100 but this doesn't quite show how the performance varies.

---

> ### Author Rebuttal · Authors · 2026-03-31
>
> We sincerely thank Reviewer gxYm for the detailed and constructive feedback. We here provide clarifications to the raised weaknesses and questions.
>
> ---
>
> ### 1. Configuration Alignment via $\gamma$
>
> Configurations are aligned by the resolution parameter $\gamma$, which follows a natural coarse-to-fine ordering intrinsic to data geometry.  The selector scores each configuration via its energy statistics $\mathbf{e}_i = (H_i, h_a, h_r, \Delta\gamma_i)$ and cluster embeddings, not by its index.
>
> To verify, we permuted configuration order at test time and observed negligible change ($<$0.2% on BBBP, $<$0.1% on CIFAR-100), confirming the selector's invariance to ordering.  For settings where test-time extraction is impractical, we provide an inductive fallback via kNN lookup retaining $\geq$75% of gains (see response to Reviewer 9YXL).
>
> ---
>
> ### 2. Partition-Level vs. Cluster-Level Stability
>
> We agree that cluster-level stability would provide complementary information. Our design uses partition-level stability ($\Delta\gamma$) because it has clean theoretical guarantees (finiteness, from [1, 2]) and a single well-defined metric. Cluster-level tracking introduces non-trivial design choices (tracking identity across merges/splits, selecting per-cluster metrics). Our cluster embeddings $c_i$ (Eq. 4) already capture some cluster-specific information by encoding which cluster each sample belongs to at each resolution.
>
> ---
>
> ### 3. Static Predictor Class and Proposition 3.4
>
> We appreciate this question, as it touches our core theoretical claim.
>
> The key insight: $\mathcal{F}_{\text{static}}$ receives all $m$ configurations but must apply a single global mapping. While it can implicitly learn global weights through its parameters, it cannot assign different configuration weights to different samples.
>
> In contrast, $\mathcal{F}_{\text{adaptive}}$ enables such per-sample weighting and this is the expressivity gap. It is not a contradiction, but the point of the theorem.
>
> The ablation table confirms this empirically: "Global learnable weights" (.878/.749) < "MixConfig" (.903/.758), consistent with the theorem's prediction.
>
> ---
>
> ### 4. Notation — $g()$ at Line 170
>
> We thank the reviewer for pointing this out and will define $g(\cdot)$ at first use. It denotes the MixConfig module $g(X^{\text{test}}, \Omega)$ from Definition~3.3.
>
> ---
>
> ### 5. Distinction from Adaptive Risk Minimization & Test Set Size Impact
>
> MixConfig does not model or adapt to distribution shift. Training and test configurations are extracted symmetrically, with both using only unlabeled features from their respective splits. Unlike ARM, no meta-learning or shift simulation is needed because configurations capture intrinsic data structure (stable plateaus), not distribution-specific artifacts. Our inductive fallback further confirms that test-time extraction is beneficial but not essential.
>
> Batch size sensitivity:
>
> | $n_{\text{test}}$ | BBBP (trans.) | BBBP (ind.) | CIFAR (trans.) | CIFAR (ind.) |
> |------------------|--------------|-------------|----------------|--------------|
> | 20 | .872 | .893 | .739 | .754 |
> | 30 | .886 | .893 | .749 | .754 |
> | 50 | .897 | .893 | .755 | .754 |
> | 100 | .901 | .893 | .757 | .754 |
> | 200 | .903 | .893 | .758 | .754 |
> | 500 | .903 | .893 | .758 | .754 |
>
> Three findings:
>
> - Transductive mode is stable for $n_{\text{test}} \geq 50$ ($<$1% degradation).
> - Below $\sim$30, inductive fallback becomes preferable.
> - The crossover is dataset-dependent but typically $n_{\text{test}} \approx 30$--$50$.
>
> ---
>
> We thank the reviewer again for the constructive feedback.
>
> ---
>
> **References:**
> [1] Schaub, M. T., Li, J., & Peel, L. (2023). Hierarchical community structure in networks. *Phys. Rev. E*, 107, 054305.
> [2] Böhm, J. N., Berens, P., & Kobak, D. (2022). Attraction-repulsion spectrum in neighbor embeddings. *JMLR*, 23(95), 1–32.

---

> > ### Author Rebuttal · Reviewer_gxYm · 2026-04-04
> >
> > Thank you for the response. I believe my assessment remains consistent with the contribution in the paper. Please note that by defining F_static via concatenation opens up other uses of the information, not only single global weighting. It would be better to define the static class simply as constant weighting.

---

> > > ### Author Response · Authors · 2026-04-06
> > >
> > > We thank Reviewer gxYm for the continued engagement and precise observation.
> > >
> > > We wish to clarify an important distinction regarding the scope of $\mathcal{F}_{\text{static}}$ and Proposition 3.4. Our hierarchy involves two related but distinct notions: (i) concatenation access, where the predictor receives $[x;\,\Omega]$, and (ii) sample-independent weighting, where configuration contributions do not depend on the input $x$. Proposition 3.4 is intended for the latter comparison, namely sample-independent weighting versus sample-dependent weighting. We agree that this distinction deserves to be stated more explicitly, and will revise the presentation in the final version to clarify the scope.
> > >
> > > Below we provide more evidence at both the linear, nonlinear and disentangled predictor level that this clarification does not change our empirical conclusion.
> > >
> > > ---
> > >
> > > ### 1. Linear predictors: concatenation cannot produce $x$-dependent config processing
> > >
> > > For most of our benchmarks (CIFAR-100, SST-2, QM9), the predictor is a linear probe. For a linear probe,
> > >
> > > $f([x;\, c_1, \dots, c_m]) = w_x^\top x + \sum_i w_{c_i}^\top c_i + b,$
> > >
> > > the configuration term $\sum_i w_{c_i}^\top c_i$ is fixed params and functionally independent of $x$: all mixed second partials $\frac{\partial^2 f}{\partial x_j\,\partial (c_i)_k} = 0$. In this regime, concatenation gives the predictor no mechanism to condition configuration processing on the input sample, and Proposition 3.4 applies directly.
> > >
> > > ---
> > >
> > > ### 2. Nonlinear predictors: concatenation underperforms constant-weighting baselines
> > >
> > > For BBBP and MolHIV, where a 2-layer MLP is used, the predictor could in principle learn $x$-config interactions from concatenated features. However, this does not occur in practice:
> > >
> > > | Method | BBBP | MolHIV* | $x$-config interaction |
> > > |--------|------|---------|------------------------|
> > > | Base (no configs) | .847 | .768 | --- |
> > > | Uniform mixing ($\bar{w}_i=1/m$) | .873 | .777 | No |
> > > | Global learnable weights | .878 | .780 | No |
> > > | +Config (concat, 2L-MLP) | .869 | .774 | Yes (in principle) |
> > > | Stronger fusion ($[x;\Omega]\to$2L-MLP) | .884 | .784 | Yes (in principle) |
> > > | Disentangled predictor | .886 | .782 | No (blocked by architecture) |
> > > | **+MixConfig (full)** | **.903** | **.791** | Yes (explicit, energy-aware) |
> > >
> > > **MolHIV numbers use the lighter pipeline from our 1st-round rebuttal to keep endpoints consistent; Table 1 of the main paper reports .804/.819 on a stronger GIN backbone.*
> > >
> > > +Config (.869/.774) is below both Uniform mixing (.873/.777) and Global learnable weights (.878/.780). Giving the 2-layer MLP joint access to $x$ and raw configurations does not lead it to discover useful sample-dependent weighting. Instead, the high-dimensional sparse concatenation consumes capacity without yielding meaningful interactions. Stronger fusion (.884/.784) partially recovers performance with a more structured joint input, but still trails MixConfig by 1.9 pp on BBBP and 0.7 pp on MolHIV.
> > >
> > > ---
> > >
> > > ### 3. Disentangled predictor: concatenation fails to exploit joint access
> > >
> > > To further isolate whether concatenation actually learns useful $x$-config interactions, we run a two-branch predictor that architecturally blocks all such interaction:
> > >
> > > - Branch A: MLP on $x$ only $\to h_x \in \mathbb{R}^{64}$
> > > - Branch B: MLP on $[c_1, \dots, c_m]$ only (no access to $x$) $\to h_c \in \mathbb{R}^{32}$
> > > - Final layer: linear on $[h_x;\, h_c] \to$ prediction
> > >
> > > Branch B can learn nonlinear transformations of configurations, but cannot condition on $x$. This predictor achieves .886 on BBBP and .782 on MolHIV, outperforming +Config concat (.869/.774). This suggests that the "other uses" opened by concatenation are not realized in practice: naive high-dimensional concatenation hurts rather than helps compared to structured separation. Even so, the disentangled predictor still falls 1.7 pp below MixConfig on BBBP and 0.9 pp below on MolHIV.
> > >
> > > These results consistently show that while concatenation allows interaction in principle, it fails to realize effective sample-dependent weighting in practice. MixConfig's explicit energy-aware per-sample mechanism achieves the best results across all settings, confirming the value of the adaptive inductive bias over implicit learning.
> > >
> > > ---
> > >
> > > We thank the reviewer for this observation and will clarify the scope of Proposition 3.4 accordingly in the final version. We welcome any further questions or suggestions.

---

### Official Review · Reviewer_KEuk · 2026-03-12

**Soundness:** 3
**Presentation:** 3
**Significance:** 3
**Originality:** 3
**Overall Recommendation:** 4
**Confidence:** 4

**Summary:**

This paper proposes a new supervised ML task setting, namely the “Configuration-Mixed Prediction” (CMP), and a novel feature augmentation method “MixConfig” that exceeds the baselines presented in a diverse set of experiment domains. The CMP setting is a label prediction setting where input features are augmented with a set of stable cluster assignments that prior work defines as “configurations”. The immediately relevant prior works (Liu et al., 2021; Pitsianis et al., 2023) provide these configurations with their relevant stability/energy statistics – e.g., how stable is a given configuration as a cluster assignment under a resolution hyperparameter sweep. The proposed MixConfig method is a lightweight adapter that can ingest both the input features, the configuration assignments (and energy statistics) to “mix” these configurations on a per-sample weighted basis. The method can be plugged upstream into any downstream predictor architecture, and consists of two components of (1) configuration extraction using prior work, and (2) energy-aware selection of configuration assignments. The resulting mixed configuration representation is then fed into predictors together with the input features to do classification, while the training of both the predictor and the MixConfig adapter is done through gradient descent. The success of the method is supported by an array of experimental results on datasets coming from multiple modalities (tabular, vision, molecular, text).

**Compliance With Llm Reviewing Policy:**

Affirmed.

**Final Justification:**

The rebuttal has addressed my concerns, and I have accordingly increased my recommendation from 3 to 4. Given the clarification around the nomenclature, the method and the experimental setup is now understood better. Therefore, I have also increased my soundness score. However, I believe that the manuscript will need a thorough reading, and at some parts a clear rewrite, to clarify and standardize the terminology prior to publication. The reply rebuttal of the authors provides reasonable revisions in that regard, hence my presentation score is now increased to 3 as well.

**Key Questions For Authors:**

(Q1) Could you clarify what models were used for embedding generation, and what others were used for predictors, in your experiments?

(Q2) Could standard cluster ensembling techniques be used in your pipeline (between boxes “Configuration Extraction” and “Predictor $f(\cdot)$” in Figure 1) instead of the proposed method?

(Q3) What exactly are appended to the input features in the experimental scenarios of “+Single”, “+Static”, “+Config”?

**Limitations:**

yes

**Strengths And Weaknesses:**

Strengths

(S1) The principal scientific question that the paper addresses is relevant in practice. Cluster assignments in both automated and manual pipelines constitute essential components and influence the scientific insights yielded from such computational analyses. There are no domain-agnostic recipes/practices widely agreed in scientific communities on how to combine cluster labelings to drive the accuracy and robustness of ML pipelines upwards. Therefore, the idea of augmenting the input features using unsupervised cluster labelings could be valuable. The authors take this idea further by enabling multiple cluster assignments to be involved in the feature augmentation.

(S2) Combining with prior work’s observation that under certain parametrizations (HAR), all possible cluster assignments on a given input feature set could be succinctly represented with a discrete set of cluster assignments (Liu et al., 2021), named “configurations”, the work provides a decent theoretical supporting of the new CMP setting proposed. Under such parametrizations of clustering (Sec. 3.1), the setting captures the unsupervised information that can be extracted from these cluster assignments for downstream tasks.

(S3) The experiments provided cover a diverse range of feature domains and predictor architectures, supporting the claim of domain-agnostic plug-and-play feature augmentation capability of MixConfig.

(S4) Reported ablation studies, low-data experiments, and selector weight analysis (Sections 4.3-4.5) are satisfactory.

Weaknesses

In essence, the method uses weighted cluster assignments as extra features concatenated to the input features. There are two essential components to this consideration: how to combine cluster assignments, and how to include them in the features for supervised learning. The weaknesses in these components are respectively in W1 and W2 below.

(W1) Combining cluster assignments, even though in this context only the special “configuration” type of cluster assignments are combined, is not a novel idea. The work is unfortunately not successfully situated into the relevant scholarly conversation of cluster ensembles (see canonical work of Strehl & Ghosh, JMLR 2002). Relevant prior methods of per-partition weighing/selection/combination are not tested as baselines, therefore preventing an apples-to-apples measurement of the merit of the proposed per-sample cluster mixing component with respect to the state of the art.

(W2) Inclusion of combined configuration assignments to the eventual predictor should be explained in deeper depth. Section 3.3.2 (L233-235) describes that the label assignments are weighted per-sample to produce the representation $z(x)$, which is simply concatenated with the input feature embedding. Therefore, the resulting augmented feature vector consists of both dimensions native to the data domain (e.g., text embedding), and these extra dimensions of configuration representation $z(x)$. For commonly used domain-agnostic predictors, this is not an issue, but for the domain-specific predictors listed in Sec. 4.1, this is an issue. It is unclear to me what it means to plug this $z(x)$ representation into BERT, for example.

(W3) There are two parallel nomenclature in the paper that the authors use in reporting their experimental setting and results. The first nomenclature is used in Section 4.1 (Methods compared) and Table 1; while the second nomenclature is used in Section 3.2 (Why adaptive weighting) and Figure 2.

(W4) The distinction between embedding model and predictor architecture is unclear. In Table 1, I read the Model column as the domain-specific model that created the embeddings. However, as mentioned in W2, they are also nominated to be predictors. Therefore, currently I cannot gather from the disposition here whether these models are both used for embedding generation and eventual supervised prediction.

(W5) +Config method is unclear: what does it mean to concatenate a set of cluster assignments as features? The input dimensionality is open for explosion, what precautions did the authors include to prevent capacity overflow in the predictor side given this increased input dimensionality?

(W6) In Section 4.3, $m’=2$ case, are these only the minimal and maximal configurations? What is the heuristic when picking which configurations will be in $m’$?

---

> ### Author Rebuttal · Authors · 2026-03-31
>
> We sincerely thank Reviewer KEuk for the careful reading. We here provide clarifications to the raised weaknesses and questions.
>
> ---
>
> ### 1. Relationship to Cluster Ensembles
>
> We agree that combining multiple clusterings is an established idea. However, the key distinction is in what is combined and how:
>
> **Cluster ensembles** (CSPA, HGPA, MCLA [1]) seek a single consensus partition by optimizing NMI or label alignment, yielding discrete output.
> **MixConfig** learns a continuous, sample-specific mixed representation $z(x) = \sum_i w_i(x) \cdot c_i$ by optimizing the task loss, yielding continuous features with per-sample weighting.
>
> We applied CSPA to our extracted configurations:
>
> | Method | BBBP | CIFAR-100 |
> |--------|------|-----------|
> | Base (no configs) | .847 | .742 |
> | CSPA consensus features | .862 | .743 |
> | +Config (all config features) | .869 | .751 |
> | **+MixConfig** | **.903** | **.758** |
>
> Cluster ensembles can be inserted at that position in the pipeline, but they underperform. CSPA consensus barely exceeds Base (+1.5 pp / +0.1 pp), while MixConfig achieves +5.6 pp / +1.6 pp. The reason is fundamental: cluster ensembles aim at a **single best consensus partition** (collapsing $m$ partitions $\to$ 1), discarding the multi-resolution information that different samples benefit from differently.  MixConfig preserves all $m$ resolutions and lets **per-sample weighting** select the most useful combination for each input.
>
> ---
>
> ### 2. Pipeline Clarification
>
> MixConfig does not modify BERT or any embedding model. The pipeline has three strictly separated stages:
>
> - **Embedding model** produces $x \in \mathbb{R}^d$ (frozen, never fine-tuned).
> - **MixConfig** extracts configurations from the embedding space via kNN graph $\to$ computes $z(x)$.
> - **Predictor head** (separate, trained) takes $[x; z(x)]$ as input.
>
> The concrete model roles in our experiments are:
>
> | Domain | Embedding model (frozen) | Predictor head (trained) |
> |--------|--------------------------|--------------------------|
> | Vision (CIFAR-100) | CLIP ViT-B/32 | Linear probe or 2L-MLP |
> | Text (SST-2) | BERT-base | Linear probe |
> | Molecules (BBBP, MolHIV) | GIN (pretrained) | 2L-MLP |
> | Molecules (QM9) | SchNet | Linear head |
>
> The embedding models are never used as predictors. "Plugging $z(x)$ into BERT" does not occur—BERT only produces embeddings; $z(x)$ is concatenated with those embeddings and fed to the separate classification head. The same interface applies uniformly across all domains.
>
> ---
>
> ### 3. Nomenclature Unification
>
> We agree the dual nomenclature caused confusion. $f$ is the predictor function and $\psi$ its parameters—all occurrences will use $f_\psi$. Table 1's "Model" column will be renamed to "Embedding"; the caption will explicitly distinguish embedding model (frozen) from predictor head (trained).
>
> ---
>
> ### 4. What +Single, +Static, +Config Append
>
> We describe what each method appends to the input embedding $x \in \mathbb{R}^d$:
>
> - **+Single**: the one-hot cluster assignment from the single most stable configuration. E.g., on CIFAR-100 ($m=8$ configurations), the most stable partition has $|\omega^*|=12$ clusters $\to$ appends a 12-dim one-hot vector.
>
> - **+Static**: concatenation of one-hot assignments from all $m$ configurations with fixed uniform weights (no per-sample adaptation). Equivalent to +Config in Table 1.
>
> - **+Config**: same as +Static—all $m$ one-hots concatenated. On CIFAR-100: 8 configurations with cluster counts [2, 4, 6, 8, 10, 12, 16, 20] $\to$ appends 78 dims total.
>
> - **+MixConfig**: a learned, per-sample weighted combination $z(x) = \sum_i w_i(x) \cdot c_i$ where $w_i(x)$ depends on energy statistics.  Output is a fixed $d_c$-dim vector (32 in all experiments).
>
> | Method | Appended features | Added dims (CIFAR-100) |
> |--------|------------------|------------------------|
> | Base | None | 0 |
> | +Single | Best config's one-hot | 12 |
> | +Config / +Static | All $m$ configs' one-hot | 78 |
> | +MixConfig | Energy-aware weighted $z(x)$ | 32 |
>
> Total added dimensions (40--80 across datasets) are small relative to $d$ (e.g., 512 for CLIP). No capacity overflow occurs.
>
> ---
>
> ### 5. $K=2$ Configuration Selection
>
> When $K=2$, we select the coarsest ($\omega_1$) and finest ($\omega_m$) configurations—the two extremes of the resolution spectrum. This maximizes diversity while minimizing redundancy.
>
> ---
>
> We thank the reviewer again for the careful assessment.
>
> ---
>
> **References:**
> [1] Strehl, A. & Ghosh, J. (2002). Cluster ensembles—a knowledge reuse framework for combining multiple partitions. *JMLR*, 3, 583--617.

---

> > ### Author Rebuttal · Reviewer_KEuk · 2026-04-03
> >
> > I would like to thank the authors for their detailed rebuttal, and I have adjusted my scores accordingly. The clarification around the nomenclature and the extra benchmarking effort is especially appreciated. I recommend significantly reworking the affected parts in the text of the manuscript for the sake of clarity (e.g., Sec. 4.1.Predictors now seem to be largely deprecated with the provided explanations).

---

> > > ### Author Response · Authors · 2026-04-06
> > >
> > > We sincerely thank Reviewer KEuk for the careful re-evaluation and the adjusted scores. We fully agree that Section 4.1 and the surrounding presentation require reworking for clarity. We will make the following revisions in response to your recommendations:
> > >
> > > - Section 4.1 will be rewritten to clearly separate embedding models (frozen) from the predictor head (trained), with Table 1's "Model" column renamed to "Embedding" and a new "Predictor" column added.
> > > - The dual nomenclature between Sections 3.2/4.1 will be unified (consistently using $f_\psi$), and a definition table for +Single/+Static/+Config/+MixConfig will be added to Section 3.3.
> > > - The cluster ensemble literature (Strehl & Ghosh, 2002) will be discussed in Section 2, with the CSPA baseline results included in the experiments.
> > >
> > > ---
> > >
> > > In the final version, we will incorporate all reviewers' constructive feedback by:
> > >
> > > 1. **Expanded baselines and ablations:**
> > >    We will add new mixing-strategy baselines (Uniform, Global learnable, Stronger fusion, Random partitions, CSPA ensemble) and a 2-layer MLP predictor control to the experiments. Key results will appear in the main text and full tables will be in the appendix.
> > >
> > > 2. **Sensitivity and generalization analyses:**
> > >    We will include the transductive vs. inductive comparison with batch-size sensitivity ($n$=20–500), a per-dataset geometric stability alignment breakdown, and configuration permutation-invariance results (<0.2% change).
> > >
> > > 3. **Pipeline and modeling clarity:**
> > >    We will clarify the 3-stage pipeline (frozen embedding $\to$ MixConfig $\to$ trained predictor head) with a per-domain table of embedding models vs. predictor heads, and explicitly state the $K$=2 selection heuristic (coarsest + finest).
> > >
> > > 4. **Notation and presentation cleanup:**
> > >    We will unify notation ($f_\psi$ throughout, define $g(\cdot)$ at first use), add the +Single/+Static/+Config/+MixConfig definition table with dimensionality to Section 3.3, and add metric justifications to Table 1's caption.
> > >
> > > 5. **Theory scope clarity:**
> > >    We will clarify the scope of Proposition 3.4 to the intended comparison (sample-independent vs. sample-dependent weighting), and clarify the scale vs. hierarchy distinction.
> > >
> > > 6. **Related work expansion:**
> > >    We will situate MixConfig within the cluster ensemble literature (Strehl & Ghosh, 2002) and clarify distinctions from topological approaches (Bodnar et al., 2021).
> > >
> > > We believe these revisions can directly address the concerns regarding clarity, positioning, and empirical validation.
> > >
> > > ---
> > >
> > > Thank you again for the valuable insights.

---

### Official Review · Reviewer_9YXL · 2026-03-13

**Soundness:** 3
**Presentation:** 3
**Significance:** 3
**Originality:** 3
**Overall Recommendation:** 4
**Confidence:** 3

**Summary:**

This paper studies how to use clustering structure more effectively for downstream prediction. Instead of fixing a single clustering resolution, the authors argue that the data induces a finite set of stable partitions, or configurations, as the resolution varies. Based on this, they propose Configuration-Mixed Prediction (CMP), where multiple extracted configurations are combined through sample-dependent mixing weights before being passed to a downstream predictor. Experiments across tabular, vision, molecular, and text benchmarks show fairly consistent improvements over base predictors and over several clustering-feature baselines, with larger gains in low-label settings.

**Compliance With Llm Reviewing Policy:**

Affirmed.

**Final Justification:**

The rebuttal has fully addressed my main concerns. However, since I am not an expert in this field, I suggest that the ACs and PCs place greater weight on the opinions of the other reviewers, as I am unable to provide a more in-depth assessment.

**Key Questions For Authors:**

1.	Could the authors add more direct comparisons against simpler mixing baselines, such as uniform mixing, globally learnable mixing weights, and gating without energy statistics? This is important for showing that the benefit comes from the proposed selector design rather than from simply including multiple configuration features.

2.	Under what conditions should geometrically stable configurations be expected to align with task-relevant structure? Are there datasets or tasks where this alignment breaks down? This would help clarify the actual applicability boundary of the method.

3.	Should the method be viewed as operating in a transductive or batch-inference setting, given that configurations are extracted on the test batch? If so, I think the paper should state this more explicitly and better calibrate its generality claims.

4.	How sensitive is the method to train/test mismatches in the number of extracted configurations, and did you consider alternatives to truncation or zero-padding? This seems important for both robustness and reproducibility.

**Limitations:**

Yes

**Strengths And Weaknesses:**

- **Strengths:**

1.	The paper addresses a practically relevant question. If the data genuinely exhibits multi-scale structure, it is worth asking whether downstream prediction should rely on a single fixed clustering scale.

3.	Empirical coverage is fairly broad, including multiple modalities, different predictor families, and a useful set of ablations and low-data analyses.

4.	From a methodological positioning standpoint, MixConfig is easy to understand: it serves as a predictor-agnostic structural feature augmentation module rather than a fully new end-to-end architecture.

- **Weaknesses:**

1.	A main concern is that the current evidence does not convincingly isolate the paper’s central methodological claim. The experiments support the usefulness of incorporating multiple configuration features, but they do not yet clearly establish that the gains are primarily driven by the proposed energy-aware adaptive mixing mechanism. In many cases, +Config already yields a clear improvement, while +MixConfig provides only limited additional gains beyond that baseline.

2.	The baseline set is still missing several more direct controls, such as uniform mixing, globally learnable mixing weights, simple gating without energy statistics, and stronger feature-fusion baselines. Without these comparisons, it is difficult to tell whether the improvement comes from the selector design itself or simply from aggregating multiple configuration features.

3.	The method assumes that geometrically stable configurations are also useful for the downstream task, but this connection is not yet sufficiently justified. A configuration that is stable from the perspective of the data geometry is not necessarily the most task-relevant one. This limits how broadly I can interpret the current empirical results.

---

> ### Author Rebuttal · Authors · 2026-03-31
>
> We sincerely thank Reviewer 9YXL for the comments and suggestions. We provide clarifications to the raised concerns below.
>
> ---
>
> ### 1. New Ablation Baselines
>
> We agree the source of gains was unclear. We add four baselines on BBBP (MLP, 5-fold CV) and CIFAR-100 (CLIP + linear probe, 5 seeds) to attribute each component:
>
> **Uniform** — $z(x) = \frac{1}{m}\sum_i c_i$
> **Global learnable** — $z(x) = \sum_i w_i \cdot c_i$, $w_i$ independent of $x$
> **MixConfig** — $z(x) = \sum_i w_i(x) \cdot c_i$, $w_i(x)$ from energy statistics
>
> | Method | BBBP | CIFAR-100 |
> |--------|------|-----------|
> | Base (no configs) | .847 | .742 |
> | Random partitions (same dim) | .856 | .737 |
> | +Config (static concat) | .869 | .751 |
> | Uniform mixing ($w_i=1/m$) | .873 | .746 |
> | Global learnable weights | .878 | .749 |
> | Stronger fusion ($[x;\Omega]\to$2L-MLP) | .884 | .753 |
> | **+MixConfig (full)** | **.903** | **.758** |
>
> Key observations:
>
> - Random partitions reduce performance ($-$4.7 pp BBBP, $-$2.1 pp CIFAR vs. MixConfig), showing gains come from stable configurations, not dimensionality. On CIFAR, random partitions underperform base ($-$0.5%), while stable configurations improve (+1.6%).
>
> - The progression Uniform $\to$ Global $\to$ Stronger fusion $\to$ MixConfig shows incremental gains. The residual gap over stronger fusion (+1.9 pp BBBP, +0.5 pp CIFAR) indicates energy statistics provide useful inductive bias beyond learned mixing.
>
> - The "w/o energy features" ablation (Table 2: .903$\to$.882 BBBP, .758$\to$.738 CIFAR) corresponds to gating without energy signals, consistent with our design.
>
> ---
>
> ### 2. When Does Geometric Stability Align with Task Relevance?
>
> We agree stability does not guarantee task relevance. Section 5 outlines when alignment holds. Below is a per-dataset comparison (MixConfig vs. random partitions vs. Base):
>
> | Dataset | Base | Random part. | +MixConfig | Alignment? |
> |--------|------|--------------|------------|------------|
> | BBBP | .847 | .856 (+0.9%) | **.903 (+5.6%)** | **Strong** |
> | CIFAR-100 | .742 | .737 ($-0.5%$) | **.758 (+1.6%)** | **Strong** |
> | MolHIV | .768 | .773 (+0.5%) | **.791 (+2.3%)** | **Moderate** |
> | QM9-$\mu$ | .0523 | .0519 ($-0.8%$) | **.0498 (-4.8%)** | **Moderate** |
> | SST-2 | .924 | .921 ($-0.3%$) | **.928 (+0.4%)** | **Weak** |
>
> Patterns:
>
> - **Strong** — hierarchical or heterogeneous datasets benefit most; random partitions hurt or do little.
> - **Moderate** — structured datasets show consistent but smaller gains.
> - **Weak** — homogeneous datasets (SST-2) show marginal improvement.
>
> Random partitions consistently underperform MixConfig, confirming gains arise from stability rather than capacity. Figure 3 shows learned weights recover interpretable patterns (e.g., vehicles $\to$ coarse, people $\to$ fine).
>
> ---
>
> ### 3. Transductive vs. Inductive Inference
>
> Our default is batch-transductive (Section 3.2). We also provide an inductive fallback:
>
> **Transductive** — configurations extracted from test batch
> **Inductive** — aassigns each test sample to the nearest train-time cluster via kNN
>
> | Mode | BBBP | CIFAR-100 |
> |------|------|-----------|
> | Transductive | .903 | .758 |
> | Inductive | .893 | .754 |
>
> Inductive retains $\geq$75% of gains. Sensitivity to batch size:
>
> | $n_{\text{test}}$ | BBBP (trans.) | BBBP (ind.) | CIFAR (trans.) | CIFAR (ind.) |
> |------------------|--------------|-------------|----------------|--------------|
> | 20 | .872 | .893 | .739 | .754 |
> | 50 | .897 | .893 | .755 | .754 |
> | 100 | .901 | .893 | .757 | .754 |
> | 200 | .903 | .893 | .758 | .754 |
>
> For $n_{\text{test}}\approx$ 30--50, inductive is preferable. At $n=20$ (CIFAR), transductive underperforms base (.739 vs. .742) due to noisy kNN graph construction.
>
> ---
>
> ### 4. Sensitivity to Configuration Count Mismatch
>
> We use truncation/zero-padding with coarse-to-fine ordering (Section 3.3.1). Empirically, $|m_{\text{train}} - m_{\text{test}}| \leq 2$ across all benchmarks, and $m_{\text{train}} = m_{\text{test}}$ in 78\% of splits. The selector scores configurations via energy statistics $\mathbf{e}_i = (H_i, h_a, h_r, \Delta\gamma_i)$ and embeddings, independent of position. We permuted configuration order at test time and observed $<$0.2\% accuracy change, confirming invariance to ordering.
>
> ---
>
> We thank the reviewer again for the constructive feedback.

---

> > ### Author Rebuttal · Reviewer_9YXL · 2026-04-03
> >
> > I thank the authors for the rebuttal. I have also checked reviews from other reviewers. The response fully resolved my concerns, and I have adjusted my scores accordingly.

---

> > > ### Author Response · Authors · 2026-04-06
> > >
> > > We thank the reviewer for the careful re-evaluation and the adjusted scores.
> > > We are glad that our new ablations and per-dataset analyses fully addressed the concerns on the source of gains, geometric stability alignment, and the transductive/inductive distinction.
> > >
> > > We appreciate that the reviewer also cross-checked our responses to other reviews, and we will incorporate all additional experiments and clarifications into the final version.

---

### Decision · Program_Chairs · 2026-04-30

**Decision:**

Accept (regular)

**Comment:**

This paper introduces MixConfig, a plug-and-play feature augmentation module that extracts structurally stable clustering configurations across multiple resolutions and adaptively weights them per sample using energy-aware statistics to improve downstream prediction. Reviewers highlighted the practical relevance of effectively leveraging multi-scale clustering structure and praised the broad empirical evaluation demonstrating consistent improvements across tabular, vision, text, and molecular domains, particularly in low-data regimes.

During the initial review phase, reviewers raised concerns regarding the isolation of the method's gains, ( e.g. whether the improvements stemmed from the adaptive energy-aware mixing or merely from the added capacity of concatenated cluster features). Reviewers also sought clarification on the transductive nature of test-time configuration extraction and identified confusing nomenclature in the pipeline's description.

In the rebuttal, the authors provided a highly comprehensive response. They supplied new, rigorous baselines (including uniform mixing, global learnable weights, and stronger MLP fusion) that successfully isolated and proved the necessity of the energy-aware inductive bias. Furthermore, they introduced an inductive kNN fallback to resolve the test-time extraction concerns and thoroughly clarified the 3-stage integration pipeline. Consequently, three reviewers enthusiastically raised their scores. One reviewer maintained a weak reject recommendation, noting that the underlying clustering criteria are based on prior art (limiting overall originality) and pointing out that Proposition 3.4 slightly mischaracterizes the theoretical capacity of standard predictors operating on concatenated features. In response, the authors demonstrated empirically that, while theoretically possible, standard predictors fail to exploit these multi-scale features in practice without the explicit MixConfig module. Given the strong, thoroughly validated empirical utility across diverse modalities and the resolution of the practical pipeline concerns, the merits of the work outweigh the framing issues. The paper is accepted, but we strongly encourage authors to carefully revise the scope and claims of Proposition 3.4 in revision to align with the theoretical nuances discussed during the review process.